# Home birth and associated factors in Nigeria: A comparative study of rural and urban settings—Analysis of national population-based data

Emmanuel O. Adewuyi[1,2,3]*, Asa Auta[4], Olumuyiwa Omonaiye[5,6], Mary I. Adewuyi[7], Victory Olutuase[8], Kazeem Adefemi[9], Olumide A. Odeyemi[3,10], Yun Zhao[1], Gizachew A. Tessema[1,2,11], Gavin Pereira[1,11,12]

1 Curtin School of Population Health, Faculty of Health Sciences, Curtin University, Perth, Western Australia, Australia, 2 Curtin Medical Research Institute, Faculty of Health Sciences, Curtin University, Perth, Western Australia, Australia, 3 Anchor University Centre for Global Health (AUCGH), Anchor University, Lagos, Nigeria, 4 Faculty of Health, Social Care and Medicine, Edge Hill University, Ormskirk, United Kingdom, 5 School of Nursing and Midwifery and Centre for Quality and Patient Safety Research in the Institute for Health Transformation, Deakin University, Geelong, Australia, 6 Deakin University Centre for Quality and Patient Safety Research–Eastern Health Partnership, Eastern Health, Box Hill, Victoria, Australia, 7 Faculty of Health, Department of Social Work, Charles Darwin University, Darwin, Northern Territory, Australia, 8 National Institute for Health Research, Clinical Research Network: East Midlands, Research and Development, Leicestershire Partnership National Health Service Trust, Leicester, United Kingdom, 9 Newfoundland & Labrador Centre for Applied Health Research, Faculty of Medicine, Memorial University of Newfoundland, St John's, Newfoundland and Labrador, Canada, 10 College of Nursing and Health Sciences, Caring Futures Institute, Adelaide, South Australia, Australia, 11 enAble Institute, Curtin University, Perth, Western Australia, Australia, 12 Faculty of Medicine, Universitas Negeri Malang, Malang, Indonesia

* emmanuel.adewuyi@curtin.edu.au

## Abstract

### Introduction

Nigeria currently has the highest maternal mortality ratio and one of the highest neonatal mortality rates worldwide. Home birth—childbirth outside health facilities, often without skilled attendance or timely access to emergency obstetric care—may contribute to these disproportionate and avoidable adverse maternal and neonatal outcomes. National estimates often mask substantial sub-national disparities. This study examines the prevalence of home birth and associated factors across national, rural, and urban settings in Nigeria.

### Methods

We analysed data from the nationally representative cross-sectional Nigeria Demographic and Health Survey 2018, guided by Andersen's Behavioural Model. Multivariable logistic regression was used to examine the associations between home birth and various predictor variables at the national level, as well as separately for rural and urban areas in Nigeria.

**Data availability statement:** The data that support the findings of this study are available from the Demographic and Health Surveys (DHS) Program, but restrictions apply to their availability. These data were used under license for the current study and are not publicly shareable. Data are, however, freely available upon reasonable request from the DHS Program (https://dhsprogram.com/data/Access-Instructions.cfm) following their approval process. The authors did not have any special access privileges, and others may access the data in the same manner.

**Funding:** EOA was supported by the National Health and Medical Research Council (NHMRC; https://www.nhmrc.gov.au/) Investigator grants (GNT2025837). The funder has no role in the conduct of this study.

**Competing interests:** The authors have declared that no competing interests exist.

## Results

Nationally, 58.1% (95% CI: 56.5, 59.7) of mothers gave birth at home, with prevalence twice as high in rural areas (72.4%, 95% CI: 70.7, 74.0) compared to urban areas (36.1%, 95% CI: 33.6, 38.7) (p<0.001). The North-West region had the highest home birth prevalence both nationally (83.6%, 95% CI: 81.5, 85.6) and in rural (89.4%, 95% CI: 87.6, 91.0), and urban (66.6%, 95% CI: 60.5, 72.2) areas (p<0.001). The South-East recorded the lowest prevalence in rural areas (16.2%, 95% CI: 11.0, 23.3), while the South-West had the lowest in urban areas (16.7%, 95% CI: 14.1, 19.7) (p<0.001). At the national level and across all settings, factors such as low maternal and husband's education, poor household wealth, fewer than eight antenatal contacts, higher birth order, Hausa-Fulani ethnicity, and limited exposure to media (radio and television) and the internet were associated with higher odds of home birth. In rural areas, additional predictors included difficulty obtaining permission, distance to health facilities, limited decision-making autonomy, and significant regional disparities, especially in the North and South-South regions. In urban areas, young maternal age, Islamic religion, financial barriers, and poor or middle household wealth were uniquely associated with higher odds of home birth.

## Conclusion

Home birth remains highly prevalent in Nigeria, particularly in rural settings and in the northern and South-South regions, where prevalence is disproportionately high. Reducing home births requires a comprehensive approach that addresses the interplay of factors identified in this study. From a social justice and health determinants perspective, these factors are interconnected and can influence both access to and use of services. In rural areas, policies should enhance women's decision-making autonomy, reduce distance barriers, and address region-specific challenges (e.g., insecurity in northern regions). In urban areas, it is essential to address financial barriers, support young mothers, and provide culturally and religiously sensitive care. Nationally, efforts should focus on improving education, expanding and strengthening antenatal care, and increasing access to media and the internet. From an equity perspective, interventions must be tailored to specific contexts to reduce unsafe home births and ensure that all mothers, regardless of location, have equitable access to skilled, respectful, and high-quality childbirth care.

## Introduction

Reducing maternal and neonatal mortality remains a global public health priority, with initiatives such as the Millennium Development Goals (MDGs) and Sustainable Development Goals (SDGs) contributing to notable progress [1,2]. Maternal mortality refers to the death of a woman from pregnancy-related causes up to 42 days postpartum, while neonatal mortality refers to the death of an infant within the first

28 days of life [3,4]. Despite the global gains, many low-resource settings continue to experience disproportionately high mortality rates, largely due to persistent inequities in the availability, accessibility, and use of quality healthcare services [2,5,6]. Nigeria exemplifies this challenge; the country currently accounts for the highest number of global maternal deaths, over 28% of all deaths, and ranks high in the absolute number of neonatal deaths [3,4,7]. According to the World Health Organisation (WHO), Nigeria's maternal mortality ratio is 993 deaths per 100,000 live births (2023 data), classified as very high, and the highest globally [3,4]. Similarly, Nigeria's neonatal mortality rate, 34 per 1,000 live births (2023 World Bank data), ranks among the highest worldwide [8]. Home births, often occurring without skilled attendance or timely access to emergency obstetric care, are a likely contributor to these unacceptably high mortalities [9–14].

In response to this ongoing challenge, the Nigerian government has introduced several initiatives aimed at increasing access to skilled care and improving maternal and newborn outcomes. These initiatives include the Midwives Service Scheme (MSS), conditional cash transfers (CCTs), and the Basic Health Care Provision Fund (BHCPF) [15–22]. The MSS expanded midwife availability in rural areas, while CCTs sought to reduce financial barriers, and the BHCPF aimed to strengthen primary healthcare delivery as the first point of contact and referral. Although these programmes yielded some benefits, their overall implementation has been constrained in several respects. The MSS faced challenges with midwife retention, limited ongoing training, inconsistent funding and variable commitment from subnational governments [15,19]. Similarly, CCT schemes were small-scale, intermittent, and in some cases withdrawn or delayed, undermining trust and continuity of care [17,18]. The BHCPF, though promising in design [20–22], has struggled with delays in fund disbursement, capacity gaps in implementation and uneven rollout. These implementation constraints may partly explain the persistence of home births, a potential contributor to Nigeria's high maternal and neonatal mortality.

Home births, defined as childbirths occurring outside health facilities without skilled attendance (e.g., obstetricians or midwives), are associated with increased maternal and neonatal risks [12–14]. The rationale for this observation is well established: the absence of skilled attendance or comprehensive emergency obstetric care in the home settings often results in complications becoming fatal, with the potential for leading to preventable maternal and neonatal deaths [12]. In contrast, health facility births attended by skilled personnel can improve outcomes through timely management of complications such as severe bleeding, infections, obstructed labour, and eclampsia—major causes of maternal and neonatal mortality [12,23–26]. Most maternal deaths occur during labour, childbirth, or the immediate postpartum period, underscoring the critical importance of skilled attendance and access to emergency obstetric care [3,4,23]. Yet in Nigeria, home births remain common [11,12,26–30], often attended by unskilled traditional birth attendants or, in some cases, without any attendant [31–33]. This situation likely reflects deep-rooted structural inequities and sociocultural practices that perpetuate high maternal and neonatal mortality in Nigeria, highlighting the need for a deeper understanding of the underlying drivers of home birth in the country.

Previous studies in Nigeria have identified low maternal education, limited household wealth, and logistical barriers as key factors associated with home birth [11,27–29,34–36]. However, most of these studies relied on pooled national data, which may obscure important sub-national differences [35]. Data disaggregation is increasingly recognised as essential for revealing inequities across geographic and socioeconomic boundaries [37,38]. For instance, both the WHO's framework for monitoring progress toward Universal Health Coverage (UHC) and the 2024 Operational Framework for Monitoring Social Determinants of Health Equity emphasise the importance of disaggregated data for assessing equity and guiding targeted interventions [39,40]. This analytical approach can strengthen efforts to track progress towards the 2030 Agenda for Sustainable Development, which aims to 'leave no one behind' [40].

Nigeria serves as a compelling setting for the use of disaggregated data due to its highly diverse population [13]. Geographic, ethnic, demographic, and socioeconomic heterogeneity in the country may compound existing health inequities, with important implications for policymaking and resource allocation. Understanding home birth patterns across subpopulations can uncover context-specific factors and provide evidence-based insights for addressing health inequities. Tailored strategies based on such evidence can better target areas of greatest need. In line with this rationale, our previous

study assessed the prevalence and factors associated with non-utilisation of healthcare facilities for childbirth in Nigeria, focusing on rural-urban differences [11]. Building on this work and recognising the need for more granular insights, the present study uses the most recent nationally representative demographic and health survey data to examine home birth prevalence and associated factors at national, rural, and urban levels. To our knowledge, this is the first study in Nigeria to adopt a comprehensive, disaggregated approach to examining home births.

To achieve our study objectives, we applied Andersen's Behavioural Model of healthcare services utilisation [41], a theory-driven framework that guided variable selection. The model posits that several factors (environmental, predisposing, enabling, and need factors) influence healthcare utilisation—in this case, the choice between home and facility-based childbirth. We also employed the social determinants of health lens to consider the broader contexts in which people are born, live, work, and age, alongside a social justice perspective that highlights structural inequities in healthcare access [40,42–44], which we applied in discussing our findings. While Andersen's model focuses on healthcare utilisation, the social determinants and social justice lenses address broader structural and systemic factors that influence access to care. This approach recognises that what appears to be an individual 'choice' may, in fact, reflect constraints in access and underlying systemic inequities.

The social justice lens, enriched by the concept of intersectionality, further examines how overlapping social identities such as race, gender, socioeconomic status, and geographic location intersect to amplify healthcare disparities and shape access to care [45]. This dual lens extends the focus beyond individual behaviours to the systemic barriers shaping maternal healthcare utilisation. Furthermore, our study offers policy-relevant recommendations aimed at empowering women, reducing home births, and improving maternal and neonatal outcomes. Our findings are expected to support Nigeria's progress toward realising SDG target 3.1 (reducing the maternal mortality ratio to fewer than 70 per 100,000 live births) and target 3.2 (ending preventable deaths of newborns and children under five) by 2030 [1,2].

## Methods

### Study setting

Nigeria is in West Africa, between 4° and 14° N latitude and 3° and 14° E longitude. It borders Benin, Niger, Chad, and Cameroon and is Africa's most populous country, with over 230 million people, 54.9% of whom live in urban areas [46]. The country has a median age of 18.1 years, indicating a substantially young population [46]. Nigeria is highly diverse, with over 374 ethnic groups and languages. The three main groups, Hausa-Fulani, Yoruba, and Igbo, and several others, make the country one of the most multilingual nations in the world [47]. Administratively, Nigeria is divided into 36 states and the Federal Capital Territory (FCT), organised into six geopolitical zones ('regions'): North-East (six states), North Central (six states and the FCT), North-West (seven states), South-South (six states), South-West (six states), and South-East (five states) [Fig 1]. Nigeria also has 774 local government areas (LGAs), with wards or enumeration areas serving as the smallest administrative units. Approximately 63% of Nigerians experience multidimensional poverty, reflecting deep socioeconomic inequalities [48].

Nigeria's healthcare system comprises both public and private sectors, operating in parallel and with limited integration. The public sector follows a three-tier structure: primary, secondary and tertiary. Primary healthcare is managed at the local government level through primary health centres, secondary care is provided by state-owned general hospitals, and tertiary care is delivered through teaching hospitals and federal medical centres offering specialised services. Despite this structure, the private sector, including formal and informal providers, plays a major role in service delivery and accounts for a considerable proportion of healthcare utilisation in Nigeria [49].

Health facility distribution in Nigeria is highly uneven across states and regions. Of the 34,423 registered facilities, 88% are primary, 12% secondary, and only 0.25% tertiary centres, with about two-thirds publicly owned and one-third private

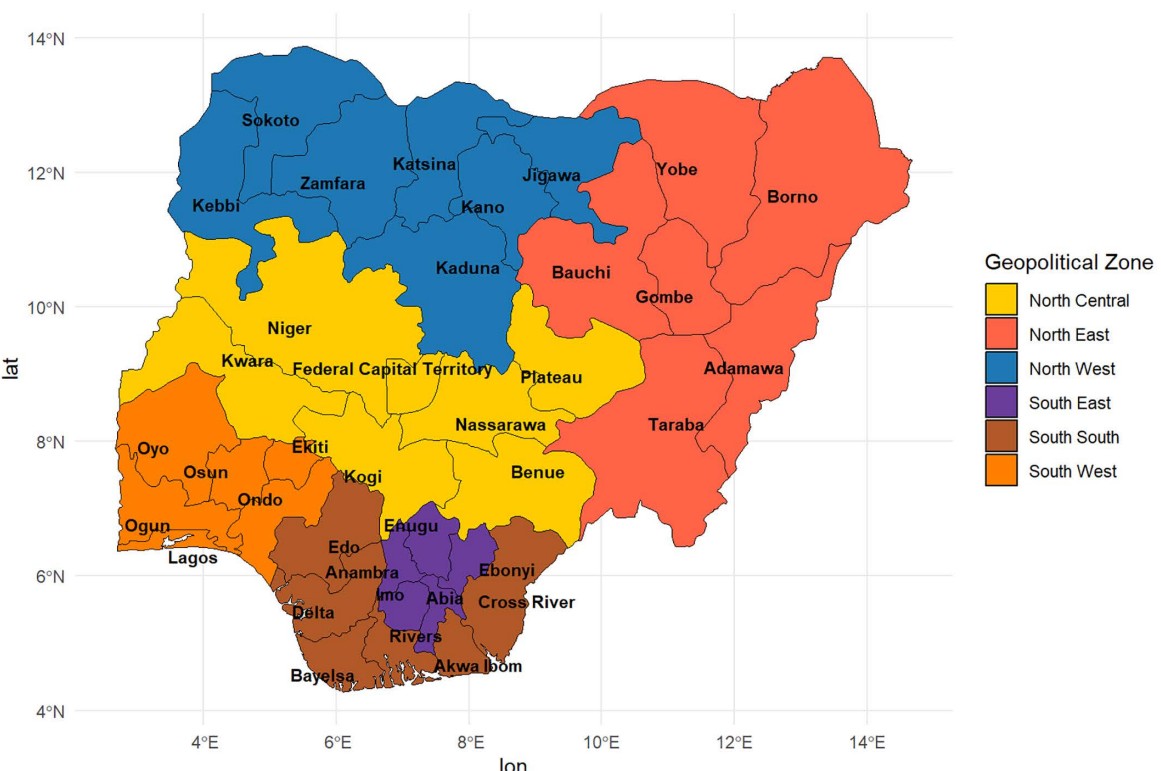

**Fig 1. Map of Nigeria showing the geo-political zones.** Note: Map of Nigeria showing states and geopolitical zones. The map was generated by the authors using R with data from the rnaturalearth and rnaturalearthdata packages (public domain). A freely accessible script for generating the map is available at https://github.com/actright1/Nigeria-Geopolitical-Map.

[50]. On average, there are 16.6 primary, 2.2 secondary, and 0.05 tertiary facilities per 100,000 population [50]; however, this figure masks sharp regional disparities. For example, the North Central has the highest density of primary facilities (23.3 per 100,000), while the South-South has the lowest (11.8 per 100,000). Secondary facilities are concentrated in the South-East (5.4 per 100,000) but remain scarce in the North-West (0.5 per 100,000). Tertiary facilities are clustered in the South-West and South-South, with a minimal presence in the North-East and North-West. Moreover, private facilities are more prevalent in southern states, whereas northern states such as Jigawa, Sokoto, and Zamfara rely more on public facilities [50].

Facility distribution also reflects a clear North–South and urban–rural divide: southern and urban areas host more private, secondary, and tertiary facilities, while northern and rural areas depend heavily on public primary centres, often under-resourced [50]. These disparities are corroborated by the Lancet Nigeria Commission, which found that facility density is consistently higher in the south than in the north, and that several northern states also scored lowest on the health facility quality index, underscoring gaps in availability, infrastructure and staffing quality [51]. The uneven distribution of healthcare facilities in Nigeria may contribute to poorer access and lower quality of care in northern and rural regions. Despite the Nigerian government's efforts to improve maternal and neonatal health in line with global health goals, through initiatives such as the MSS, CCTs, and BHCPF [15–18,20–22], mentioned earlier, poor maternal and neonatal health outcomes continue to persist. Home birth is a potential contributor, and national data may obscure within-population disparities [34,51,52]; hence, the approach in this study.

## Data source and sampling design

We utilised data from the cross-sectional Nigerian Demographic and Health Survey (NDHS) 2018. The NDHS is a nationally representative survey conducted periodically since its inception in 1990 [13]. The survey employs a rigorously tested and validated methodology. The 2018 version was implemented by the Nigerian National Population Commission in collaboration with international partners and technical oversight from ICF International. The survey provides information on key demographic and health indicators, including fertility, family planning, maternal and child health, and health service utilisation. The NDHS 2018 used a stratified two-stage cluster sampling method to select 42,000 households from 1,400 clusters [13]. Data collection for the survey was conducted between 14 August and 29 December 2018 using standardised, validated questionnaires administered by trained interviewers [13]. The completed survey covered 40,427 households within 1,389 clusters. A total of 41,821 mothers aged 15–49 participated (16,984 from urban and 24,837 from rural areas). The overall response rate was high at 99.3%, with urban and rural response rates of 99.2% and 99.4%, respectively.

In this study, we analysed a weighted representative sample of 21,512 mothers—8,467 from urban areas and 13,055 from rural areas—who had a live birth in the five years before the survey [13]. These mothers provided complete data on the place of childbirth for their most recent live birth. We used the Children Recode (KR) dataset, restricted to the most recent live birth per mother. This dataset includes detailed records on pregnancy, childbirth, postnatal care, and child health, together with maternal information, fully anonymised to protect personally identifiable data. Documentation on the NDHS 2018 methodology is publicly available [13]. Ethical approval for the NDHS 2018 survey was granted by the Nigerian National Health Research Ethics Committee, and informed consent was obtained from participants aged 18 and older, while parental consent was required for those under 18. The dataset can be accessed online at https://dhsprogram.com/data/available-datasets.cfm, upon approval from the DHS Program [13]. As our study involved secondary analysis of fully de-identified publicly available data, no additional ethics approval was required.

## Study factors

**Outcome variable.** The primary outcome variable was 'home birth' or 'home delivery', representing non-utilisation of a health facility or institution for childbirth. This outcome was derived from the 'place of delivery' variable, which was categorised into two groups: 'home birth' and 'health facility birth.' Based on our data, the 'home birth' category follows the practice in previous studies [10,11], comprising births occurring at the 'respondent's home' and 'another home'. Conversely, health facility births encompassed childbirths occurring in public and private healthcare facilities. Public facilities included government hospitals, health centres, health posts, and other public sector institutions, while private facilities comprised private hospitals, clinics, and other privately operated healthcare providers/facilities.

**Explanatory variables.** We adopted Andersen's Behavioural Model as the conceptual framework and utilised it for selecting explanatory variables, a well-established approach in healthcare services utilisation research [11,34,35,53]. The model categorises factors influencing healthcare utilisation into four key domains: environmental factors (external conditions, including broader contextual influences), predisposing factors (demographic and social characteristics shaping healthcare-seeking behaviour), enabling factors (resources that facilitate or hinder access to care), and need factors (individual perceptions and evaluations of health needs) [41]. Using this framework, we examined how various factors are associated with having a birth at home rather than in a healthcare facility. The selection of explanatory variables was also guided by a review of previous research and the availability of information in our dataset [10,11,34,35,53]. These variables were grouped into four broad categories (Fig 2), categorised or re-categorised as presented in Table 1.

(i) External environmental factors included respondents' place of residence (urban or rural) and regions (northern geopolitical zones: North-West, North-Central and North-East; and southern geopolitical zones: South-South, South-West,

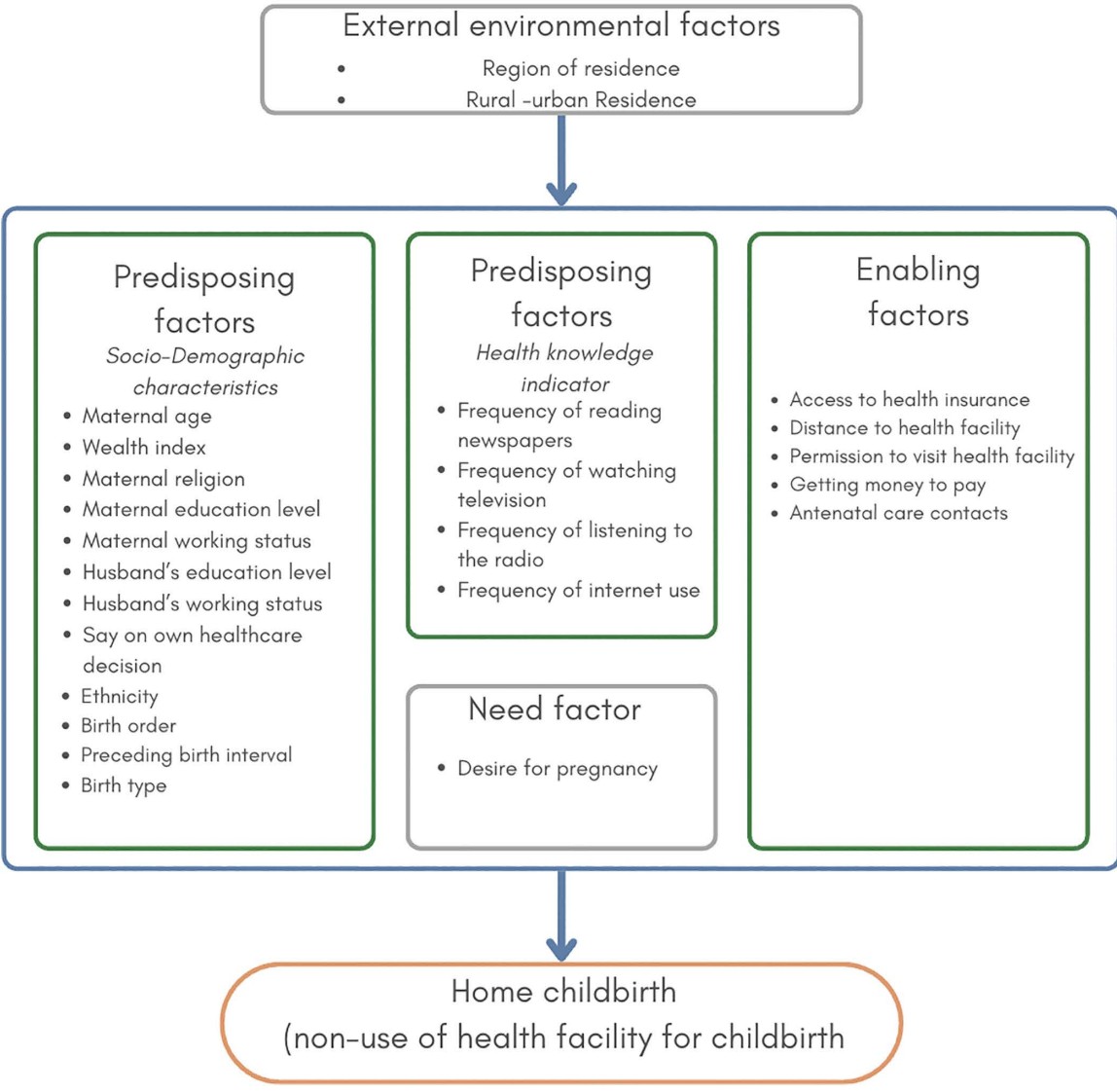

**Fig 2. Theoretical framework for home birth in Nigeria.**

and South-East) classified separately (Figs 1 and 2). We followed the same classification of these variables as in the original DHS 2018 data and report [13].

(ii) Predisposing factors were classified into socio-demographic characteristics and health knowledge indicators. Socio-demographic characteristics included maternal age, categorised into seven groups: 15–19, 20–24, 25–29, 30–34, 35–39, 40–44, and 45–49 years [13,34,52]. Household wealth, a composite measure of socioeconomic status derived from principal component analysis of household assets, was reclassified from the original five NDHS 2018 categories into three groups for this study: poor (poorest and poorer), middle (middle), and rich (richer and richest) [10,13,35]. Other socio-demographic factors include religion, education level, working status of the mother and her husband/partner, decision-making autonomy (say on own health), ethnicity, birth order, preceding birth interval, and birth type [13]. Health knowledge indicators include frequency of exposure to newspapers, television, radio, and the

**Table 1. Sample characteristics and home birth in Nigeria by rural and urban residences, NDHS 2018.**

| Factors | Nigeria (overall) | | | Rural Nigeria | | | Urban Nigeria | | |
|---|---|---|---|---|---|---|---|---|---|
| | Weighted sample (N=21,512) (%) [a] | Home birth [b] | | Weighted sample (N=13045) (%) [a] | Home birth [b] | | Weighted sample (N=8467) (%) [a] | Home birth [b] | |
| | | % (95%CI) | P-Value | | % (95%CI) | P-Value | | % (95%CI) | P-Value |
| **Place of birth** | | | <0.001** | | | <0.001** | | | <0.001** |
| **Home** | 12497 | 58.1 (56.5, 59.7) | | 9441 | 72.4 (70.7, 74.0) | | 3055 | 36.1 (33.6, 38.7) | |
| **Health facility** | 9015 | 41.9 (40.3, 43.5) [#] | | 3604 | 27.6 (26.0, 29.3) [#] | | 5412 | 63.9 (61.3, 66.4) [#] | |
| *External environmental factors* | | | | | | | | | |
| **Region of residence** | | | <0.001** | | | <0.001** | | | <0.001** |
| North-Central | 3005 (14.0) | 49.2 (45.8, 52.6) | | 2066 (15.8) | 55.1 (51.3, 58.9) | | 939 (11.1) | 36.2 (29.4, 43.6) | |
| North-East | 3857 (17.9) | 73.3 (69.9, 76.4) | | 2958 (22.7) | 79.2 (75.2, 82.6) | | 899 (10.6) | 53.9 (47.0, 60.8) | |
| North-West | 7642 (35.5) | 83.6 (81.5, 85.6) | | 5703 (43.7) | 89.4 (87.6, 91.0) | | 1939 (22.9) | 66.6 (60.5, 72.2) | |
| South-East | 2118 (9.8) | 18.4 (15.4, 21.9) | | 571 (4.4) | 16.2 (11.0, 23.3) | | 1546 (18.3) | 19.2 (15.7, 23.4) | |
| South-South | 1885 (8.8) | 45.5 (41.6, 49.4) | | 1089 (8.3) | 55.8 (51.0, 60.5) | | 796 (9.4) | 31.4 (26.2, 37.0) | |
| South-West | 3005 (14) | 18.4 (16.0, 21.1) | | 658 (5.0) | 24.6 (18.9, 31.2) | | 2347 (27.7) | 16.7 (14.1, 19.7) | |
| **Rural-urban residence** | | | <0.001** | | | | | | |
| Rural | 13045 (60.6) | 72.4 (70.7, 74.0) | | | | | | | |
| Urban | 8467 (39.4) | 36.1 (33.6, 38.7) | | | | | | | |
| *Predisposing factors (Socio-demographic characteristics)* | | | | | | | | | |
| **Maternal education level** | | | <0.001** | | | <0.001** | | | <0.001** |
| Higher | 1893 (8.8) | 11.1 (9.4, 13.1) | | 410 (3.1) | 15.4 (12.1, 19.4) | | 1483 (17.5) | 9.9 (8.0, 12.2) | |
| Secondary | 6720 (31.2) | 33.2 (31.3, 35.1) | | 2780 (21.3) | 43.1 (40.1, 46.0) | | 3940 (46.5) | 26.2 (23.9, 28.6) | |
| Primary | 3183 (14.8) | 57.0 (54.7, 59.4) | | 1917 (14.7) | 63.9 (61.2, 66.5) | | 1266 (14.9) | 46.6 (42.7, 50.6) | |
| None | 9716 (45.2) | 84.8 (83.5, 86.1) | | 7938 (60.8) | 87.6 (86.2, 89.0) | | 1778 (21.0) | 72.3 (68.6, 75.8) | |
| **Maternal working status** | | | <0.001** | | | <0.001** | | | <0.001** |
| Not working | 6864 (31.9) | 70.6 (68.6, 72.6) | | 4642 (35.6) | 82.3 (80.4, 84.0) | | 2222 (26.2) | 46.2 (42.2, 50.3) | |
| Working | 14648 (68.1) | 52.2 (50.5, 54.0) | | 8403 (64.4) | 66.9 (64.8, 68.9) | | 6245 (73.8) | 32.5 (30.1, 35.0) | |
| **Husband/partner's education level** | | | <0.001** | | | <0.001** | | | <0.001** |
| Higher | 3095 (15.5) | 27.9 (25.4, 30.7) | | 1003 (8.2) | 41.6 (37.8, 45.6) | | 2092 (27.0) | 21.4 (18.3, 24.8) | |
| Secondary | 6749 (33.8) | 40.9 (39.0, 49.2) | | 3339 (27.4) | 52.2 (49.5, 54.8) | | 3410 (44.1) | 29.9 (27.2, 32.8) | |
| Primary | 2786 (14.0) | 60 (57.4, 62.6) | | 1783 (14.6) | 71.0 (68.1, 73.1) | | 1002 (13.0) | 40.5 (36.2, 45.0) | |
| None | 7314 (36.7) | 87.4 (85.9, 88.7) | | 6081 (49.8) | 90.3 (88.9, 91.6) | | 1233 (15.9) | 73.0 (67.9, 77.5) | |
| **Husband/partner's working status** | | | <0.001** | | | <0.001** | | | 0.008** |
| Not working | 678 (3.4) | 73.8 (68.0, 78.9) | | 454 (3.7) | 85.6 (80.7, 89.4) | | 224 (2.9) | 49.8 (39.2, 60.3) | |
| Working | 19529 (96.6) | 58.2 (56.5, 59.9) | | 11944 (96.3) | 72.6 (70.9, 74.3) | | 7586 (97.1) | 35.5 (32.8, 38.2) | |
| **Wealth index** | | | <0.001** | | | <0.001** | | | <0.001** |
| Rich | 7639 (35.5) | 28.0 (26.0, 30.0) | | 1920 (14.7) | 36.4 (32.9, 40.0) | | 5719 (67.5) | 25.2 (22.9, 27.6) | |
| Middle | 4364 (20.3) | 57.5 (54.6, 60.3) | | 2699 (20.7) | 60.9 (57.1, 64.5) | | 1665 (19.7) | 52.0 (48.1, 55.8) | |
| Poor | 9509 (44.2) | 82.6 (81.1, 84.0) | | 8426 (64.6,) | 84.3 (82.7, 85.7) | | 1083 (12.8) | 69.4 (63.5, 74.7) | |
| **Maternal age (years)** | | | <0.001** | | | 0.008** | | | <0.001** |
| 15-19 | 1205 (5.6) | 72.0 (68.6, 75.1) | | 963 (7.4) | 77.1 (73.7, 80.2) | | 241 (2.8) | 51.4 (42.2, 60.4) | |
| 20-24 | 4134 (19.2) | 63.9 (61.5, 66.3) | | 2849 (21.8) | 73.4 (70.7, 75.9) | | 1285 (15.2) | 43.1 (39.1, 47.2) | |
| 25-29 | 5574 (25.9) | 56.8 (54.7, 58.9) | | 3334 (25.6) | 71.9 (69.7, 74.1) | | 2240 (26.5) | 34.4 (31.2, 37.7) | |
| 30-34 | 4669 (21.7) | 53.3 (51.0, 55.6) | | 2537 (19.5) | 70.9 (68.2, 73.4) | | 2132 (25.2) | 32.4 (29.4, 35.6) | |
| 35-39 | 3550 (16.5) | 52.8 (50.4, 55.2) | | 1928 (14.8) | 70.1 (67.4, 72.7) | | 1623 (19.2) | 32.3 (28.9, 35.9) | |

*(Continued)*

| Factors | Nigeria (overall) | | | Rural Nigeria | | | Urban Nigeria | | |
|---|---|---|---|---|---|---|---|---|---|
| | Weighted sample (N=21,512) (%) [a] | Home birth [b] | | Weighted sample (N=13045) (%) [a] | Home birth [b] | | Weighted sample (N=8467) (%) [a] | Home birth [b] | |
| | | % (95%CI) | P-Value | | % (95%CI) | P-Value | | % (95%CI) | P-Value |
| 40-44 | 1690 (7.9) | 59.7 (56.8, 62.6) | | 1014 (7.8) | 73.8 (70.6, 76.8) | | 676 (8.0) | 38.6 (33.3, 44.1) | |
| 45-49 | 690 (3.2) | 64.2 (59.9, 68.2) | | 419 (3.2) | 74.4 (69.7, 78.6) | | 271 (3.2) | 48.3 (41.0, 55.7) | |
| **Maternal religion** | | | <0.001** | | | <0.001** | | | <0.001** |
| Christianity | 8055 (37.4) | 31.9 (29.8, 34.0) | | 3889 (29.8) | 45.2 (42.1, 48.3) | | 4166 (49.2) | 19.5 (17.5, 21.6) | |
| Traditional/others | 117 (0.5) | 70.7 (56.8, 81.5) | | 81 (0.6) | 78.1 (64.0, 87.8) | | 36 (4.0) | 53.9 (35.8, 71.0) | |
| Islam | 13340 (62.0) | 73.8 (72.0, 75.6) | | 9075 (69.6) | 84.0 (82.1, 85.6) | | 4265 (50.4) | 72.8 (69.3, 76.1) | |
| **Ethnicity** | | | <0.001** | | | <0.001** | | | <0.001** |
| Hausa-Fulani | 9625 (44.7) | 82.7 (80.9, 84.3) | | 7148 (54.8) | 88.5 (86.8, 89.9) | | 2478 (29.3) | 66.1 (61.6, 70.3) | |
| Yoruba | 2549 (11.9) | 18.5 (16.1, 21.1) | | 521 (4.0) | 18.9 (15.1, 23.5) | | 2028 (24.0) | 18.3 (15.6, 21.5) | |
| Igbo | 2727 (12.7) | 17.6 (15.1, 20.5) | | 732 (5.6) | 19.6 (14.6, 25.7) | | 1995 (23.6) | 4.4 (3.6, 5.4) | |
| Others | 6610 (30.7) | 54.2 (51.9, 56.5) | | 4644 (35.6) | 61.9 (59.2, 64.6) | | 1966 (23.2) | 36.1 (32.4, 39.9) | |
| **Birth order** | | | <0.001** | | | <0.001** | | | <0.001** |
| 1 | 3681 (17.1) | 46.3 (43.9, 48.7) | | 2101 (16.1) | 61.8 (59.1, 64.5) | | 1581 (18.7) | 25.6 (22.5, 29.0) | |
| 2 - 3 | 7120 (33.1) | 51.2 (49.0, 53.4) | | 3967 (30.4) | 68.4 (65.8, 70.9) | | 3153 (37.2) | 29.6 (26.8, 32.7) | |
| ≥ 4 | 10710 (49.8) | 66.7 (65.1, 68.3) | | 6977 (53.5) | 77.8 (76.2, 79.3) | | 3733 (44.1) | 46.0 (43.0, 49.0) | |
| **Preceding birth interval** | | | 0.465 | | | 0.650 | | | <0.001** |
| < 24 months | 3657 (20.6) | 61.2 (58.9, 63.6) | | 2264 (20.7) | 74.9 (72.3, 77.4) | | 1392 (20.3) | 39.0 (35.1, 43.1) | |
| ≥ 24 months | 14136 (79.4) | 60.5 (58.8, 62.1) | | 8658 (79.3) | 74.4 (72.6, 76.1) | | 5478 (79.7) | 38.4 (35.7, 41.2) | |
| **Birth type** | | | <0.001** | | | <0.001** | | | <0.001** |
| Multiple | 421 (2.0) | 42.1 (37.0, 47.4) | | 246 (1.9) | 58.1 (51.4, 64.6) | | 175 (2.1) | 19.6 (13.7, 27.1) | |
| Single | 21091 (98.0) | 58.4 (56.8, 60.0) | | 12799 (98.1) | 72.6 (70.9, 74.3) | | 8292 (97.1) | 36.4 (33.9, 39.1) | |
| **Final say on own health** | | | <0.001** | | | <0.001** | | | <0.001** |
| Respondent alone | 1292 (9.5) | 43.9 (40.2, 47.6) | | 860 (6.9) | 62.5 (58.1, 66.8) | | 1068 (13.8) | 28.9 (24.3, 34.0) | |
| Respondent and husband/ partner | 6201 (30.6) | 38.9 (36.7, 41.2) | | 2901 (23.3) | 55.4 (52.4, 58.3) | | 3300 (42.2) | 24.5 (21.7, 27.5) | |
| Husband/partner/someone else/other | 12142 (59.9) | 71.3 (69.6, 73.0) | | 8682 (69.8) | 80.1 (78.4, 81.8) | | 3460 (44.2) | 49.1 (45.6, 52.6) | |
| *Predisposing factors (Health knowledge indicators)* | | | | | | | | | |
| **Frequency of reading newspaper/magazine** | | | <0.001** | | | <0.001** | | | <0.001** |
| Not all | 19002 (88.3) | 63.0 (61.4, 64.5) | | 12276 (94.1) | 74.8 (73.2, 76.4) | | 6726 (79.4) | 41.2 (38.5, 44.1) | |
| <once a week | 1809 (8.4) | 22.8 (20.4, 25.5) | | 552 (4.2) | 36.9 (32.6, 41.6) | | 1257 (14.8) | 16.6 (14.0, 19.7) | |
| ≥ Once a week | 701 (3.3) | 17.3 (14.2, 20.9) | | 216 (1.7) | 22.5 (17.0, 29.2) | | 484 (5.7) | 14.9 (11.3, 19.4) | |
| **Frequency of listening to radio** | | | <0.001** | | | <0.001** | | | <0.001** |
| Not all | 10150 (47.2) | 72.9 (71.2, 74.5) | | 7503 (57.5) | 80.9 (79.2, 82.5) | | 2647 (31.3) | 50.1 (46.4, 53.9) | |
| <once a week | 5210 (24.2) | 48.7 (46.4, 51.0) | | 2783 (21.3) | 63.7 (60.9, 66.4) | | 2427 (28.7) | 31.4 (28.4, 34.5) | |
| ≥ once a week | 6152 (28.6) | 41.7 (39.4, 44.0) | | 2758 (21.1) | 57.9 (54.9, 60.9) | | 3393 (40.1) | 28.5 (25.6, 31.6) | |
| **Frequency of watching television** | | | <0.001** | | | <0.001** | | | <0.001** |
| Not all | 12041 (56.0) | 77.6 (76.1, 79.0) | | 9514 (72.9) | 82.0 (80.5, 83.5) | | 2526 (29.8) | 60.8 (56.9, 64.5) | |
| <once a week | 3695 (17.2) | 42.6 (40.3, 44.9) | | 1775 (13.6) | 53.0 (50.1, 56.0) | | 1919 (22.7) | 32.9 (29.7, 36.3) | |
| ≥ Once a week | 5776 (26.9) | 27.4 (25.3, 29.6) | | 1755 (13.5) | 39.7 (35.6, 43.9) | | 4022 (47.5) | 22.1 (19.8, 24.5) | |

*(Continued)*

**Table 1.** (Continued)

| Factors | Nigeria (overall) | | | Rural Nigeria | | | Urban Nigeria | | |
|---|---|---|---|---|---|---|---|---|---|
| | Weighted sample (N=21,512) (%) a | Home birth b | | Weighted sample (N=13045) (%) a | Home birth b | | Weighted sample (N=8467) (%) a | Home birth b | |
| | | % (95%CI) | P-Value | | % (95%CI) | P-Value | | % (95%CI) | P-Value |
| **Frequency of Internet use** | | | <0.001** | | | <0.001** | | | <0.001** |
| Not all | 19419 (90.3) | 63.1 (61.6, 64.6) | | 12698 (97.3) | 74.0 (72.3, 75.6) | | 6722 (79.4) | 42.6 (40.0, 45.3) | |
| <once a week | 353 (1.6) | 13.0 (9.2, 18.1) | | 68 (0.5) | 19.0 (11.4, 30.2) | | 286 (3.4) | 11.6 (7.5, 17.6) | |
| ≥ Once a week | 1740 (8.1) | 11.1 (9.1, 13.6) | | 280 (2.1) | 12.7 (9.1, 17.6) | | 1460 (17.2) | 10.8 (8.5, 13.6) | |
| *Enabling factors* | | | | | | | | | |
| **Access to health insurance** | | | <0.001** | | | 0.021* | | | <0.001** |
| No | 21035 (97.8) | 58.8 (57.2, 60.4) | | 12900 (98.9) | 72.6 (70.8, 74.2) | | 8135 (96.1) | 36.9 (34.4, 39.5) | |
| Yes | 477 (2.2) | 27.8 (20.1, 37.1) | | 145 (1.1) | 55.6 (39.4, 70.6) | | 332 (3.9) | 15.6 (10.8, 22.0) | |
| **Distance to health facility** | | | <0.001** | | | <0.001** | | | 0.280 |
| Big problem | 6032 (28.0) | 68.6 (66.1, 71.0) | | 4577 (35.1) | 78.2 (75.9, 80.3) | | 1455 (17.2) | 38.6 (33.3, 44.3) | |
| Not a big problem | 15480 (72.0) | 54.0 (52.2, 55.8) | | 8468 (64.9) | 69.3 (67.2, 71.2) | | 7012 (82.8) | 35. 6 (33.0, 38.2) | |
| **Permission to visit health facility** | | | <0.001** | | | <0.001** | | | 0.087 |
| Big problem | 2549 (11.8) | 71.2 (67.8, 74.3) | | 1876 (14.4) | 81.9 (78.7, 84.7) | | 673 (7.9) | 41.2 (34.9, 47.8) | |
| Not a big problem | 18963 (88.2) | 56.3 (54.7, 58.0) | | 11169 (85.6) | 70.8 (69.0, 72.5) | | 7794 (92.1) | 35.6 (33.1, 38.3) | |
| **Getting money for health services** | | | <0.001** | | | <0.001** | | | <0.001** |
| Big problem | 10375 (48.2) | 64.8 (62.9, 66.7) | | 7134 (54.7) | 74.7 (72.7, 76.7) | | 3240 (38.3) | 43 (39.5, 46.5) | |
| Not a big problem | 11137 (51.8) | 51.8 (49.9, 53.7) | | 5911 (45.3) | 69.5 (67.4, 71.6) | | 5227 (61.7) | 31.8 (29.3, 34.5) | |
| **ANC contacts** | | | <0.001** | | | <0.001** | | | <0.001** |
| 8 or more contacts | 4225 (20.0) | 19.3 (17.6, 21.1) | | 1333 (10.3) | 27.1 (23.9, 30.6) | | 2891 (35.2) | 15.7 (13.9, 17.8) | |
| 7 or less contacts | 16943 (80.0) | 68.5 (66.9, 70.0) | | 11629 (89.7) | 77.8 (76.1, 79.4) | | 5314 (64.8) | 48.1 (45.0, 51.2) | |
| *Need factor* | | | | | | | | | |
| **Desire for pregnancy** | | | <0.001** | | | <0.001** | | | <0.001** |
| Then | 18969 (88.2) | 59.8 (58.1, 61.4) | | 11765 (90.2) | 74.0 (72.2, 75.7) | | 7204 (85.1) | 36.7 (34, 39.4) | |
| Later | 1845 (8.6) | 45.7(42.8, 48.6) | | 924 (7.1) | 57.0 (53.1, 60.7) | | 921 (10.9) | 34.4 (30.6, 38.5) | |
| No more | 698 (3.2) | 44.4 (39.8, 49.2) | | 356 (2.7) | 60.0 (54.6, 65.2) | | 342 (4.0) | 28.2 (22.0, 35.4) | |

#: Relates to birth in health facilities only,

*Significant at 5% level,

**Significant at 1% level,

a Weighted sample size and percentages.

b Weighted percentage of home birth.

N=sample size (weighted).

NDHS, Nigeria Demographic and Health Survey.

internet, all of which were categorised as 'not at all', 'less than once a week', and 'at least once a week'. The variables were categorised as described in Table 1.

(iii) Enabling factors encompass key aspects influencing healthcare access, such as health insurance, antenatal care contacts, financial capacity to pay for services, distance to facility, and difficulties in obtaining permission to seek care. The categories of these variables are shown in Table 1.

(iv) Need factors focused on pregnancy intention or desire for pregnancy, representing the mother's intention at the time of conception.

Further details on the classification and operationalisation of these variables are in Table 1.

## Statistical analysis

We assessed both unadjusted (univariable) and adjusted (multivariable) relationships between home birth and explanatory variables using a two-step approach. First, we performed analyses using aggregated data for the overall Nigerian population; second, we disaggregated our data and conducted separate analyses for urban and rural settings. In each setting (overall, rural and urban areas), we estimated the proportion of home births, along with the corresponding 95% confidence intervals (CIs), using frequency tabulation. Unadjusted associations between home birth and each explanatory factor were evaluated using Chi-square tests, and only variables with statistically significant associations ($p < 0.05$) were advanced to the multivariable analyses. Analyses were restricted to observations with complete data; cases with 'missing' or 'don't know' responses were excluded.

For the multivariable analyses, we built logistic regression models for all four variable categories and employed a backward elimination strategy to systematically remove non-significant variables, retaining only those that were statistically significant at 5% level. Before fitting the models, we assessed multicollinearity using Variance Inflation Factors (VIFs) and tolerance values to assess the independence of predictor variables. The VIFs values <10 and tolerance values >0.1 were considered acceptable thresholds [54]. Variable selection for model building was guided by both theoretical relevance (based on the literature) and statistical evidence (significance in our univariable analysis). All statistical analyses and data management were conducted in IBM SPSS Statistics for Windows (Version 21.0). For both univariable and multivariable analyses, we used the Complex Samples module of SPSS to account for sampling weights, stratification, and clustering in the NDHS 2018 design. This approach ensures that our estimates accurately reflect the survey's complex sampling structure, as well as that standard errors and significance tests reflect the true population variability [55].

To ensure the robustness of the logistic regression model, we applied a model evaluation strategy appropriate for complex survey data. Model fit was assessed using design-adjusted Wald F-statistics and pseudo R-squared values (Cox & Snell, Nagelkerke, McFadden) [56]. Although pseudo $R^2$ values do not represent explained variance in the same way as in linear regression, they provide a useful comparative metric for assessing model adequacy. The Hosmer–Lemeshow test was not used, as it assumes simple random sampling [57] and is not supported in SPSS Complex Samples Logistic Regression. We further evaluated the final parsimonious model by examining potential confounders and factors previously reported to be associated with home birth.

## Results

### Sample characteristics for the overall Nigerian data

The overall weighted sample consisted of 21,512 mothers aged 15–49 years. Regionally, the North-West had the highest representation at 35.5%, while the South-South had the lowest at 8.8% [Table 1]. Nearly half of the mothers (45.2%) had no formal education, while 8.8% had higher education. In terms of age, the cumulative percentage reaches just over 50% in the 25–29 age group. Using the grouped data formula, the estimated median age was 29.9 years. Teenagers comprised 5.6% of the sample, while most mothers were between 20 and 34 years (66.8%), highlighting a predominantly young-middle-aged population. Additionally, 68.1% of the mothers were working, 35.5% were in rich households, while 44.2% were in poor households.

### Sample characteristics for rural and urban residences

Rural-urban data disaggregation revealed notable differences (Table 1). Of the total sample, 60.6% (13,045) resided in rural areas and 39.4% (8,467) in urban areas. In rural areas, a higher proportion of mothers had no formal education

(60.8%) compared to urban mothers (21.0%). Working status also differed, with 64.4% of rural mothers working compared with 73.8% of urban mothers. However, urban residents typically had higher educational attainment (17.5% with higher education) than their rural counterparts (3.21%), suggesting variations in the type and quality of work. In urban areas, 67.5% of mothers were in the rich category, and only 12.8% were in the poor category. In contrast, just 14.7% of rural mothers were rich, and 64.6% were in the poor category.

### Prevalence of home birth in Nigeria by rural and urban residence

Overall, 58.1% (95% CI: 56.5, 59.7) of mothers in Nigeria gave birth at home, while 41.9% (95% CI: 40.3, 43.5, p<0.001) delivered at a health facility (Fig 3). Home birth prevalence was twice as high in rural (72.4% [95% CI: 70.7, 74.0]) as in urban areas (36.1% [95% CI: 33.6, 38.7], p<0.001). At the national level, the North-West had the highest home birth prevalence (83.6% [95% CI: 81.5, 85.6]), while the South-East (18.4%; 95% CI: 15.4, 21.9) and South-West (18.4%; 95% CI: 16.0, 21.1, p<0.001) had the lowest.

Considering rural–urban residence-specific differences, the South-East had the lowest home birth prevalence in rural areas (16.2%; 95% CI: 11.0, 23.3), while the South-West had the lowest in urban areas (16.7%; 95% CI: 14.1, 19.7) (Fig 4). Conversely, the North-West recorded the highest prevalence, with 89.4% (95% CI: 87.6, 91.0) in rural areas and 66.6% (95% CI: 60.5, 72.2, p<0.001) in urban areas. Across all regions, except the South-East, rural areas consistently reported a higher home birth prevalence than urban areas. Notably, in the South-East region, home birth was slightly higher (but not statistically significant) in urban settings (19.2%; 95% CI: 15.7, 23.4) than in rural areas (16.2%; 95% CI: 11.0, 23.3) with overlapping CIs. Overall, the southern regions demonstrated lower home birth prevalence than the northern regions (Fig 4). In all settings, the Hausa-Fulani ethnic group had the highest home birth prevalence (82.7% in the overall, 88.5% in rural and 66.1% in urban areas) compared to other ethnic groups. Also, mothers who had no education, identified as Muslim, had fewer than eight antenatal care contacts, were teenagers, or belonged to the low wealth index category had higher home birth prevalence.

### Multicollinearity and model fit results

We assessed multicollinearity and model fit in examining predictors of home birth. All VIFs were below five, and all tolerance values exceeded 0.1, indicating no significant multicollinearity among the independent variables [54]. These results suggest that the predictors in the final model are sufficiently distinct and that the estimated coefficients are stable and

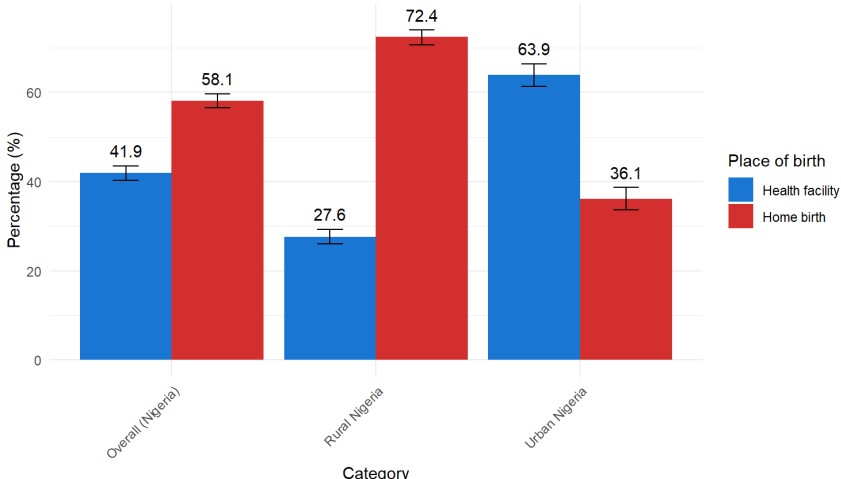

**Fig 3. Home and health facility birth prevalence in Nigeria by national, rural and urban contexts.**

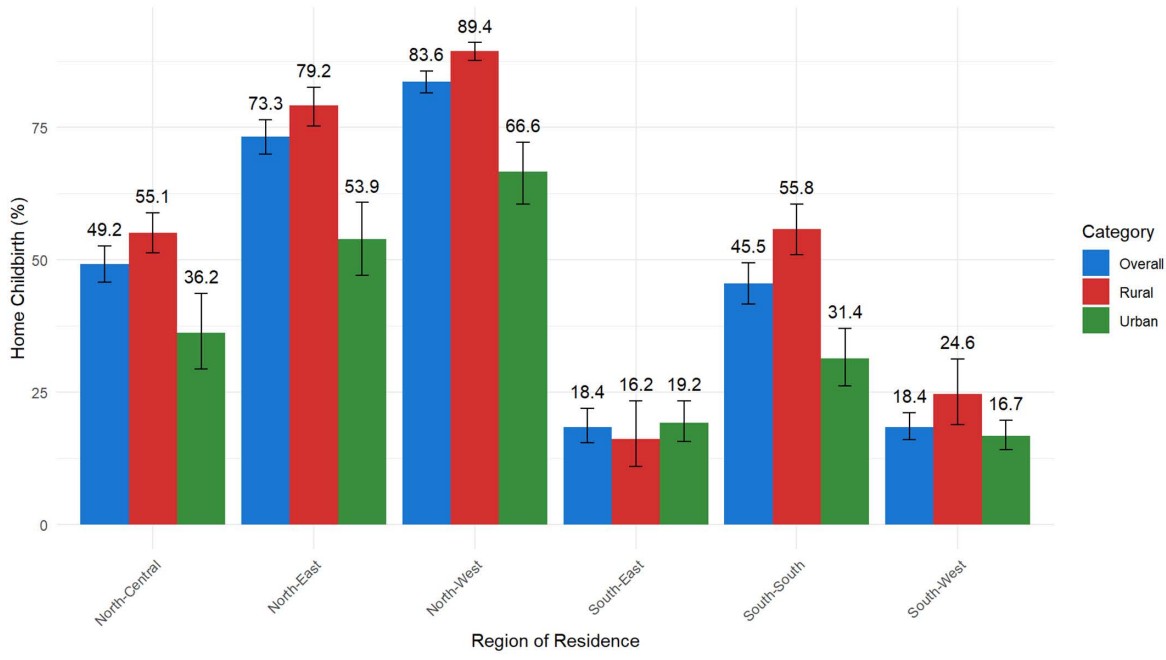

**Fig 4. Home birth in Nigeria's geopolitical zones by national, rural and urban residences.**

interpretable. All models also demonstrated statistically significant overall fit (p<0.001), suggesting that the predictors collectively contributed meaningfully to the likelihood of home birth in each setting. For instance, in the national model, the overall test was significant (F(36, 1278) = 67.511, $p<0.001$), indicating strong model performance. Similarly, both the rural (F (30, 746) = 52.10, $p<0.001$) and urban (F (35, 504) = 23.68, $p<0.001$) models showed significance. The smaller Wald F-statistic values in the rural (52.10) and urban (23.68) models may partly reflect reduced sample size and increased estimation variability in those subpopulations. Regardless, all models remained statistically significant. Pseudo R-squared values supported the Wald F-statistics. For the national model, the Cox & Snell R² was 0.395, the Nagelkerke R² was 0.533, and the McFadden R² was 0.372. The urban and rural models also demonstrated acceptable model performance. The pseudo R-squared values for the residences fall within similar ranges (though slightly lower for rural and urban compared to the national). Notably, McFadden R² values of 0.2–0.4 are typically regarded as indicative of good fit in logistic regression [58]. Together, these results suggest the final models achieved a good balance of explanatory power and parsimony, providing confidence in interpreting the associations.

## Factors associated with home birth in the overall Nigerian population

In the multivariable analyses, region of residence (an environmental factor) was significantly associated with home birth. Specifically, mothers residing in the North-East region (AOR=1.87; 95%CI: 1.34, 2.61), North-West (AOR=3.30; 95%CI: 2.36, 4.64), and South-South (AOR=4.54; 95%CI: 3.21, 6.43) had significantly higher odds of home birth compared to the South-West (Table 2). Nine sociodemographic characteristics (predisposing factors) were significantly associated with home birth in the overall Nigerian population (Table 2). For instance, mothers with no education (AOR=3.18; 95% CI: 2.35, 4.30), and those with primary education (AOR=2.41; 95% CI: 1.80, 3.23), or secondary education (AOR=1.87; 95% CI: 1.43, 2.45) had greater odds of home birth in comparison with higher education. A similar pattern of results was observed for the husband/partner's education level (Table 2).

**Table 2. Factors associated with home birth in Nigeria across the overall, rural and urban areas.**

| Factors | Nigeria (Overall) | | | Rural Nigeria | | | Urban Nigeria | | |
|---|---|---|---|---|---|---|---|---|---|
| | AOR | 95%CI | P | AOR | 95%CI | P | AOR | 95%CI | P |
| *External environmental factor* | | | | | | | | | |
| **Region** | | | <0.001** | | | <0.001** | | | <0.001** |
| North-Central | 1.28 | 0.93, 1.76 | 0.136 | 1.77 | 1.07, 2.94 | 0.027* | 1.07 | 0.72, 1.58 | 0.750 |
| North-East | 1.87 | 1.34, 2.61 | <0.001** | 3.11 | 1.87, 5.19 | <0.001** | 1.30 | 0.83, 2.02 | 0.252 |
| North-West | 3.30 | 2.36, 4.64 | <0.001** | 5.83 | 3.43, 9.90 | <0.001** | 2.09 | 1.38, 3.17 | 0.001** |
| South-East | 1.44 | 0.87, 2.40 | 0.158 | 1.63 | 0.75, 3.54 | 0.220 | 1.25 | 0.66, 2.38 | 0.488 |
| South-South | 4.54 | 3.21, 6.43 | <0.001** | 7.82 | 4.55, 13.44 | <0.001** | 2.71 | 1.74, 4.20 | <0.001** |
| South-West | 1.00 | Reference | | 1.00 | Reference | | 1.00 | Reference | |
| *Predisposing factors (Socio-demographic characteristics)* | | | | | | | | | |
| **Maternal education level** | | | <0.001** | | | <0.001** | | | <0.001** |
| No education | 3.18 | 2.35, 4.30 | <0.001** | 3.46 | 2.33, 5.12 | <0.001** | 3.10 | 2.03, 4.75 | <0.001** |
| Primary | 2.41 | 1.80, 3.23 | <0.001** | 2.47 | 1.69, 3.60 | <0.001** | 2.67 | 1.78, 4.01 | <0.001** |
| Secondary | 1.87 | 1.43, 2.45 | <0.001** | 1.97 | 1.38, 2.79 | <0.001** | 1.88 | 1.30, 2.70 | 0.001** |
| Higher | 1.00 | Reference | | 1.00 | Reference | | 1.00 | Reference | |
| **Husband's education level** | | | <0.001** | | | <0.001** | | | 0.059 |
| No Education | 2.26 | 1.84, 2.77 | <0.001** | 2.77 | 2.18, 3.53 | <0.001** | 1.57 | 1.11, 2.21 | 0.010* |
| Primary | 1.55 | 1.29, 1.86 | <0.001** | 1.76 | 1.39, 2.21 | <0.001** | 1.34 | 1.01, 1.77 | 0.040* |
| Secondary | 1.30 | 1.11, 1.52 | 0.001** | 1.34 | 1.10, 1.63 | 0.004** | 1.24 | 0.98, 1.57 | 0.080 |
| Higher | 1.00 | Reference | | 1.00 | Reference | | 1.00 | Reference | |
| **Wealth index** | | | <0.001** | | | <0.001** | | | <0.001** |
| Poor | 2.19 | 1.85, 2.59 | <0.001** | 1.80 | 1.43, 2.27 | <0.001** | 2.17 | 1.64, 2.87 | <0.001** |
| Middle | 1.35 | 1.17, 1.54 | <0.001** | 1.20 | 0.98, 1.46 | 0.073 | 1.46 | 1.22, 1.75 | <0.001** |
| Rich | 1.00 | Reference | | 1.00 | Reference | | 1.00 | Reference | |
| **Birth type** | | | <0.001** | | | 0.002** | | | <0.001** |
| Single | 2.27 | 1.65, 3.12 | <0.001** | 1.94 | 1.29, 2.91 | 0.002** | 3.07 | 1.71, 5.51 | <0.001** |
| Multiple | 1.00 | Reference | | 1.00 | Reference | | 1.00 | Reference | |
| **Birth order** | | | <0.001** | | | <0.001** | | | 0.008** |
| 2 - 3 vs. 1 | 1.51 | 1.28, 1.78 | <0.001** | 1.58 | 1.32, 1.89 | <0.001** | 1.42 | 1.07, 1.89 | 0.002** |
| ≥ 4 | 1.75 | 1.45, 2.12 | 0.026* | 1.65 | 1.41, 1.94 | 0.500 | 1.68 | 1.21, 2.34 | 0.081 |
| 1 | 1.00 | Reference | | 1.00 | Reference | | 1.00 | Reference | |
| **Ethnicity** | | | <0.001** | | | <0.001** | | | <0.001** |
| Hausa-Fulani | 1.53 | 1.26, 1.87 | <0.001** | 1.52 | 1.16, 2.00 | 0.002** | 1.63 | 1.25, 2.13 | <0.001** |
| Yoruba | 1.02 | 0.72, 1.45 | 0.909 | 0.86 | 0.51, 1.46 | 0.574 | 0.99 | 0.65, 1.50 | 0.950 |
| Igbo | 0.51 | 0.35, 0.75 | 0.001** | 0.48 | 0.29, 0.79 | 0.004** | 0.54 | 0.33, 0.88 | 0.013* |
| Others | 1.00 | Reference | | 1.00 | Reference | | 1.00 | Reference | |
| **Final say on own health** | | | | | | 0.004** | | | |
| Respondent alone | | | | 0.73 | 0.59, 0.91 | 0.005** | | | |
| Respondent and spouse | | | | 0.83 | 0.72, 0.96 | 0.014* | | | |
| Spouse alone/someone else/others | | | | 1.00 | Reference | | | | |
| **Maternal religion** | | | 0.005** | | | | | | 0.001** |
| Traditionalist/others | 1.45 | 0.88, 2.40 | 0.719 | | | | 2.05 | 0.97, 4.36 | 0.496 |
| Islam | 1.32 | 1.10, 1.58 | 0.003** | | | | 1.55 | 1.19, 2.02 | 0.001** |
| Christianity | 1.00 | Reference | | | | | 1.00 | Reference | |
| **Maternal age** | | | 0.021* | | | | | | 0.043* |
| 15-19 | 1.493 | 1.09, 2.05 | 0.013* | | | | 2.186 | 1.25, 3.82 | 0.006** |

*(Continued)*

**Table 2.** (Continued)

| Factors | Nigeria (Overall) | | | Rural Nigeria | | | Urban Nigeria | | |
|---|---|---|---|---|---|---|---|---|---|
| | AOR | 95%CI | P | AOR | 95%CI | P | AOR | 95%CI | P |
| 20-24 | 1.498 | 1.17, 1.92 | 0.001** | | | | 1.786 | 1.21, 2.64 | 0.004** |
| 25-29 | 1.429 | 1.14, 1.79 | 0.002** | | | | 1.652 | 1.15, 2.36 | 0.006** |
| 30-34 | 1.264 | 1.01, 1.58 | 0.040* | | | | 1.515 | 1.06, 2.16 | 0.021* |
| 35-39 | 1.171 | 0.94, 1.47 | 0.168 | | | | 1.317 | 0.92, 1.87 | 0.127 |
| 40-44 | 1.220 | 0.92, 1.61 | 0.160 | | | | 1.421 | 0.91, 2.23 | 0.125 |
| 45-49 | 1.00 | Reference | | | | | 1.00 | Reference | |
| *Predisposing factors (Health knowledge indicators)* | | | | | | | | | |
| **Frequency of watching TV** | | | <0.001** | | | 0.003** | | | <0.001** |
| Not at all | 1.42 | 1.22, 1.64 | <0.001** | 1.26 | 1.00, 1.59 | 0.046* | 1.56 | 1.28, 1.89 | <0.001** |
| <once a week | 1.10 | 0.94, 1.28 | 0.239 | 0.93 | 0.74, 1.16 | 0.523 | 1.23 | 1.00, 1.51 | 0.049* |
| ≥ once a week | 1.00 | Reference | | 1.00 | Reference | | 1.00 | Reference | |
| **Frequency of listening to radio** | | | <0.001** | | | 0.003** | | | 0.051 |
| Not at all | 1.36 | 1.18, 1.56 | <0.001** | 1.35 | 1.14, 1.60 | 0.001** | 1.30 | 1.04, 1.62 | 0.019* |
| <once a week | 1.14 | 1.00, 1.31 | 0.054 | 1.23 | 1.02, 1.49 | 0.033* | 1.08 | 0.89, 1.31 | 0.432 |
| ≥ once a week | 1.00 | Reference | | 1.00 | Reference | | 1.00 | Reference | |
| **Frequency of using the Internet** | | | 0.001** | | | 0.001** | | | 0.010* |
| Not at all | 1.51 | 1.12, 2.02 | 0.006** | 2.53 | 1.54, 4.15 | <0.001** | 2.41 | 1.33, 4.36 | 0.002** |
| <Once a week | 0.74 | 0.45, 1.22 | 0.238 | 1.76 | 0.68, 4.60 | 0.246 | 1.81 | 1.00, 3.28 | 0.051 |
| ≥ Once a week | 1.00 | Reference | | 1.00 | Reference | | 1.00 | Reference | |
| *Enabling factors* | | | | | | | | | |
| **Antenatal contacts** | | | <0.001** | | | <0.001** | | | <0.001** |
| < 7 ANC contacts (underuse) | 2.35 | 2.02, 2.74 | <0.001** | 3.03 | 2.43, 3.78 | <0.001** | 1.91 | 1.57, 2.32 | <0.001** |
| ≥8 ANC contacts (use) | 1.00 | Reference | | 1.00 | Reference | | 1.00 | Reference | |
| **Permission to visit health facility** | | | 0.019* | | | 0.001** | | | |
| Big problem | 1.22 | 1.03, 1.44 | 0.019* | 1.45 | 1.15, 1.82 | 0.001** | | | |
| Not a big problem | 1.00 | Reference | | 1.00 | Reference | | | | |
| **Distance to health facility** | | | <0.001** | | | <0.001** | | | |
| Big problem | 1.29 | 1.14, 1.46 | <0.001** | 1.37 | 1.18, 1.60 | <0.001** | | | |
| Not a big problem | 1.00 | Reference | | 1.00 | Reference | | | | |
| **Getting money for health services** | | | | | | | | | 0.006** |
| Big problem | | | | | | | 1.25 | 1.07, 1.47 | 0.006** |
| Not a big problem | | | | | | | 1.00 | Reference | |

AOR: Adjusted odds ratio,

CI: confidence interval.

P: P-value.

*Significant at 5% level,

**Significant at 1% level.

We examined factors associated with home birth in Nigeria using multivariable binary logistic regression. Cells are left blank where variables were not significant in the setting (national, rural or urban area).

Factors assessed but not significant at the national (Nigeria overall) level: final say on own health, getting money for health services, frequency of reading newspapers, health insurance coverage, maternal working status, husband working status, and desire for pregnancy, preceding birth interval.

Factors assessed but not significant in rural residence: maternal religion, maternal age, getting money for health services, frequency of reading newspapers, health insurance coverage, maternal working status, husband working status, desire for pregnancy, and preceding birth interval.

Factors assessed but not significant in urban residence: Final say on own health, permission to visit health facility, distance to health facility, frequency of reading newspaper, health insurance coverage, maternal working status, husband working status, desire for pregnancy, and preceding birth interval.

Additionally, mothers under 35 had higher odds of home birth compared to older age groups. Notably, in a step-wise pattern, mothers from poor households (AOR = 2.19; 95% CI: 1.85–2.59) and those in the middle wealth category (AOR = 1.35; 95% CI: 1.17–1.54) had greater odds of a home birth compared to those from rich households. In addition, giving birth to a singleton, rather than multiples, was associated with increased odds of home birth. Higher birth order, compared to first births, and identifying as Muslim, compared to Christian, were also associated with elevated odds. Belonging to the Hausa-Fulani ethnic group increased the odds of home birth, while Igbo ethnicity was associated with substantially lower odds (Table 2).

Among the health knowledge indicators (predisposing factors), mothers who reported not watching television at all had higher odds of home birth (AOR = 1.42; 95% CI: 1.22, 1.64) compared to those who watched at least once weekly. Similarly, those who did not listen to the radio at all had increased odds of home birth (AOR = 1.36; 95% CI: 1.18, 1.56). Not using the internet at all also increased the odds of home birth (AOR = 1.51; 95% CI: 1.12–2.02).

Lastly, among the enabling factors, mothers who reported challenges in accessing health services recorded higher odds of home birth (Table 2). Specifically, those who found obtaining permission to visit a facility problematic (AOR = 1.22; 95% CI: 1.03, 1.44) and those who described distance to a facility as a 'big problem' (AOR = 1.29; 95% CI: 1.14, 1.46) had increased odds of home birth. Finally, ANC contact was inversely related to home birth: mothers who had fewer than seven ANC contacts had markedly higher odds compared to those with eight or more contacts.

### Similarities in factors associated with home birth between rural and urban settings

In both rural and urban settings, mothers with no education had significantly higher odds of home birth compared to those with higher education. The effect was slightly stronger in rural areas than in urban areas, but remained comparable (Table 2). Similarly, lower educational attainment at the primary and secondary levels was associated with increased odds of home birth in both settings. In both contexts, mothers from poor households also had greater odds of home birth compared to those from rich households (Table 2). A lack of partner's education was associated with higher odds of home birth in both settings, with a slightly stronger effect in rural areas (AOR = 2.77; 95% CI: 2.18–3.53) than in urban areas (AOR = 1.57; 95% CI: 1.11–2.21). Mothers who recorded fewer than seven antenatal contacts had higher odds of home birth in both settings, with a stronger association in rural areas (AOR = 3.03; 95% CI: 2.43–3.78) than in urban areas (AOR = 1.91; 95% CI: 1.57–2.32). Additionally, higher birth order, Hausa-Fulani ethnicity, singleton births, and limited media exposure (TV, radio, and internet) were common factors associated with home birth across both rural and urban areas (Table 2).

### Differences in factors associated with home birth between rural and urban settings

Notable differences emerged when comparing rural and urban residences. First, the odds of home birth were significantly higher in four rural regions: North-East, North-Central, North-West, and South-South, compared to the South-West, and by extension, the South-East. That is, in rural Nigeria, all the northern regions (North-Central, North-West and North-East), and only one southern region (South-South) had higher odds of home birth. Conversely, in urban areas, higher odds were observed in only two regions: one from the northern area (North-West) and one from the southern area (South-South). Regional disparities were more pronounced in rural areas. For example, the North-West region was associated with higher odds of home birth in both settings, but the effect was over twice as strong in rural (AOR = 5.83; 95% CI: 3.43–9.90) than urban areas (AOR = 2.09; 95% CI: 1.38–3.17). Similarly, in the South-South region, rural mothers had nearly three-fold higher odds (AOR = 7.82; 95% CI: 4.55–13.44) compared to urban counterparts (AOR = 2.71; 95% CI: 1.74–4.20).

Secondly, certain enabling factors were uniquely associated with rural areas (not significant in urban areas). Mothers who had difficulty obtaining permission to visit a health facility had higher odds of home birth (AOR = 1.45; 95% CI: 1.15–1.82), as did those reporting distance as a major barrier (AOR = 1.37; 95% CI: 1.18–1.60). Home birth odds were also higher among mothers not involved in healthcare decision-making. Specifically, those deciding alone or jointly with

partners had lower odds of home birth, a pattern observed only in rural areas (Table 2). Thirdly, in urban areas, maternal age, religion, and financial access were uniquely associated with home birth. Younger mothers, particularly those aged 15–19 (AOR = 2.19; 95% CI: 1.25–3.82) and 20–24 (AOR = 1.79; 95% CI: 1.21–2.65), had higher odds compared to older groups. Religion was significant only in urban settings; Muslim mothers had higher odds than Christians (AOR = 1.55; 95% CI: 1.19–2.02). Financial constraints also mattered in urban areas, where mothers in the middle wealth index (AOR = 1.46; 95% CI: 1.22–1.75) and those who reported difficulty affording care (AOR = 1.25; 95% CI: 1.07–1.47) had higher odds of home birth.

## Discussion

We present a comprehensive assessment of the prevalence and factors associated with home birth in Nigeria, stratified by rural and urban residence. Our findings reveal a high national prevalence, with 58% of mothers giving birth at home. We observed marked differences by rural–urban residence, region, health knowledge indicators, socio-demographic characteristics, and enabling factors. The rural–urban divide is striking: nearly three-quarters (72%) of rural mothers gave birth at home, compared to just over one-third (36%) of urban mothers, indicating a widespread challenge, with rural areas recording more than twice the urban prevalence. Although current figures represent a modest decline from the 2013 estimates (63% nationally, 78% rural, and 38% urban) [11,30], the prevalence remains one of the highest globally. The national estimate exceeds that of many African countries and is more than twice the global average of 28% [59–62], reflecting only marginal improvements despite interventions implemented over the past decades [15–18,20–22]. These reductions remain insufficient to meet global targets for skilled birth attendance, a key indicator for achieving SDG 3.1 [1,2]. The continued high prevalence is concerning, given that home births in Nigeria typically occur without skilled attendance, or, in some cases, any assistance at all [31,32].

Beyond national and rural–urban patterns, substantial regional variation further highlights the persistence of home births in Nigeria. For instance, we found prevalence ranging from 89% in the rural North-West to 16% in the rural South-East. High prevalence in areas such as the rural North-West (89%) suggests that many mothers are compelled to give birth at home due to constrained options, including inadequate infrastructure, shortages of skilled personnel, insecurity, restrictive sociocultural norms, and limited empowerment of women to have a say (autonomy) in matters of their own health [63]. Even in urban areas, two-thirds (67%) of mothers in the urban North-West gave birth at home, indicating that residence in urban settings alone does not ensure lower home birth prevalence. This finding suggests that the urban North-West, while slightly better, shares many of the same barriers as rural areas [63]. Notably, fewer than 20% of health facilities in Nigeria are equipped to provide emergency obstetric care [64], which may further limit women's ability to give birth in a facility. These findings underscore the urgent need to strengthen health systems and expand access to comprehensive emergency obstetric services to reduce home births.

However, the availability of health services alone does not guarantee improved maternal or perinatal outcomes [65]. The global shift towards prioritising quality of care highlights the need to ensure that services are not only available but meet quality standards capable of improving outcomes [14,65]. Yet even high-quality services are insufficient if systemic barriers prevent women from accessing them. Addressing systemic barriers requires a social justice perspective, which considers the structural and societal factors that influence women's ability to benefit from care [40,42–44]. A social justice approach goes beyond service provision to address structural inequities, such as poverty, gender disadvantage, and regional underdevelopment, which often marginalise mothers and limit their access to safe and equitable childbirth care. This approach involves ensuring fair access, empowering women through education and autonomy, and addressing the unequal distribution of resources [40,42–44].

An important observation in our study is that the South-East region deviated from the typical rural–urban pattern, with home birth slightly higher in urban (19%) than rural areas (16%). Although marginal, this observation suggests relatively uniform prevalence across settings in a region with the lowest national home birth (tied with the South-West). This finding

aligns with a previous study [11], which reported a similar rural–urban pattern. One explanation lies in the South-East's unique socio-geographic characteristics. Evidence indicates a rural–urban symbiosis, characterised by strong kinship ties and frequent mobility [66], which may reduce gaps in infrastructure, health literacy and service access. Unlike more isolated or under-resourced rural areas in other regions, rural communities in the South East benefit from communal development and diaspora investments [66]. Hence, they may have similar proximity to health facilities and possess comparable resources for utilising care as their urban counterparts, potentially explaining the observed parity.

Following multivariable analyses, home birth was significantly associated with several predisposing factors, including maternal and husband/partner education, household wealth, ethnicity, birth order, birth type, and media (radio, television) and internet exposure. These findings align with Andersen's predisposing domain [41], capturing demographic and social characteristics shaping service utilisation. Associations of home births with these factors remained significant nationally and in stratified urban–rural analyses, suggesting they had a consistent influence on childbirth location across Nigeria. Our findings support evidence from studies in Nigeria [10,11,27–29], other parts of Africa [59,67], and globally [68]. While some factors reflect individual or household decision-making, collectively current findings are better interpreted through broader social determinants of health and intersectionality, shaped by structural inequities that constrain equitable access to facility-based childbirth [42,43,45]. These interrelated factors may also reflect cumulative disadvantage: women with limited education, low wealth, high parity, and poor health information can be systematically disadvantaged by structural conditions rather than choice. Addressing these patterns requires equity-focused policies that go beyond service provision to tackle root causes such as low education, poverty, and rural underdevelopment in Nigeria.

Specifically, maternal education demonstrated a clear dose–response association, with the odds of home birth declining as educational attainment increased across national, rural, and urban settings. This finding underscores education's pivotal role as a social determinant of health and, within Andersen's model [41], a predisposing factor empowering mothers through greater health literacy, decision-making autonomy, and financial resources for institution-based childbirth. Moreover, education shapes health-seeking behaviour [69] and can influence the intersecting effects of gender, poverty, and geography on access to maternal healthcare. Recognised as a fundamental human right [69,70], education remains inequitable in Nigeria, where over 10 million school-age children—about 60% girls—are out of school [71]. This inequity highlights a social justice concern and a crucial policy entry point. Investing in girls' education, in line with SDG 4 [72], is a pathway not only to improved maternal health but also an urgent social justice imperative. The consistency of the findings across all settings in Nigeria parallels previous studies [10,11,27], reinforcing education's cross-cutting influence on maternal health outcomes.

Further emphasising the pivotal role of educational attainment, mothers whose husbands were educated had lower odds of home birth, in all residences, largely mirroring the dose–response pattern observed for maternal education. Our finding aligns with previous studies conducted nationally [27], across rural and urban areas [11], and among young mothers in Nigeria [10]. Limited partner education may reduce encouragement or support for facility-based care and reflect lower household socioeconomic status, reinforcing persistent inequalities. Consistent with this interpretation, our findings highlight socioeconomic inequalities, as mothers from poor households recorded higher odds of home birth than their wealthier counterparts across all settings. Within Andersen's model [41], household wealth functions as a predisposing factor, a marker of long-term social positioning that shapes healthcare opportunities and access. Our findings are consistent with previous studies that demonstrate a strong association between low wealth status and reduced maternal healthcare utilisation [10,11,29,73].

The higher home birth odds observed among the Hausa-Fulani [41] likely reflect structural disadvantages linked to ethnicity across national, rural, and urban settings. Many Hausa-Fulani communities, particularly pastoralists or nomads, reside in or move through rural and remote areas, often with limited infrastructure and health services [74]. The finding highlights how intersecting factors such as ethnicity, mobility, and geography can restrict access to and utilisation of healthcare services. Other predisposing factors, such as higher birth order, multiple births, and low media exposure (radio

and television), implicate cultural, medical, and informational underpinnings. The findings are consistent with previous studies [11,27]. Higher-parity mothers may normalise home birth due to previous safe birth experiences, while multiple births may signal a higher risk of complications, necessitating facility-based childbirth. Similarly, poor media access can limit exposure to vital health information, with potential consequences for low utilisation of maternal health services [75].

Notably, mothers without internet use had higher odds of home birth nationally and across rural and urban areas. Internet access, a 'super social determinant of health,' influences employment, education, and healthcare [76,77]. In Andersen's model [41], we categorise internet use as a predisposing (knowledge) factor, though it could also function as an enabling factor, with potential for facilitating access to health information and services. Limited use reflects broader inequities, including education and income constraints [78]. Policies promoting digital inclusion within maternal health programs could improve equity and reduce home births [78]. However, uneven access to smartphones, electricity, and networks, especially in rural areas, risks widening inequalities [79], underscoring the need to pair digital inclusion with infrastructure improvements.

Shifting focus to enabling factors, ANC illustrates their potential through its association with lower odds of home birth [41], with mothers having fewer than seven contacts showing higher odds across all settings. Beyond clinical care, ANC fosters trust in the health system, supports birth preparedness, and encourages facility-based childbirth [80,81]. The stronger rural effect suggests sociocultural and infrastructural barriers play an important role. For example, difficulty obtaining permission to visit facilities and distance to care (both enabling factors) were significant in rural but not urban areas, potentially amplifying ANC's influence on childbirth location. These findings highlight how rural-linked economic disadvantage, sociocultural practices, and structural barriers intersect to increase home births and maternal health disparities in Nigeria [11,41,50,51].

In stratified analyses, higher odds of home birth persist in the rural North-East, North-Central, North-West, and South-South, potentially echoing enduring geographical inequities and the amplifying effect of rurality through limited infrastructure, workforce shortages, and inadequate institutional presence [11,41,50,51]. Regional disparities are shaped by context-specific challenges, including poverty, insecurity, and weak healthcare coverage in the North, and underinvestment and environmental degradation in the South-South region [51,82–84]. Beyond structural constraints, rural mothers face access-related barriers rooted in gender and power dynamics, with difficulties in obtaining permission and physical distance limiting maternal autonomy and mobility. Women's participation in healthcare decision-making, independently or jointly with partners, was associated with lower odds of home birth in rural areas, highlighting the greater influence of decision-making autonomy in these settings. These findings support interventions addressing structural inequities and restrictive gender norms, while promoting partner-inclusive approaches to reduce home births in rural areas [85].

In urban areas, financial constraints, an enabling factor [41], remained significant despite expected advantages like job opportunities and service availability. This result aligns with our wealth index findings, including higher home birth odds among middle-wealth women, not seen in rural areas. Economic barriers affect both poor and middle-class households, suggesting rising costs, a shrinking middle class, and urban poverty, where disadvantage and spatial exclusion limit access to quality maternity services [13,33,73]. Lastly, mothers professing Islam, in urban areas, had higher odds of home birth than Christian or other mothers, likely reflecting cultural, religious, and social determinants that influence maternal healthcare use [86,87]. These findings highlight the need for culturally sensitive, faith-informed maternal health services that respect beliefs while promoting safe childbirth [88].

Finally, it is noteworthy that our findings indicate both higher parity and younger age as predictors of home birth, which may seem contradictory. However, the results show that age and parity exert independent effects potentially through different mechanisms, with their influence varying by context (parity across rural and urban settings, age primarily in urban areas). These differences highlight the need for interventions tailored to the specific vulnerabilities of each group [10,41].

## Policy and programmatic implications

Our study highlights the need for targeted, equity-oriented policies to address home births in Nigeria, framed through Andersen's model and a social justice perspective. This approach recognises that addressing predisposing, enabling, and environmental barriers requires interventions that are both feasible and responsive to systemic inequities. First, our findings implicate maternal and husband education as critical socio-demographic predisposing factors. Hence, it is crucial to prioritise policies that promote educational opportunities, particularly in rural, all northern and South-South regions, where home birth prevalence and odds were highest. Expanding access to quality education for girls is a critical first step. Improving maternal health literacy is also essential, enabling women to understand birth complications and make informed healthcare decisions. From a social justice perspective, empowering girls and women through formal education and health literacy addresses long-standing systemic inequities and strengthens their capacity to make autonomous health choices [89]. Community-driven initiatives, such as women's support groups, mobile health messaging, and the engagement of community health volunteers, can complement formal education by providing culturally appropriate health education and encouraging shifts toward skilled birth attendance.

Second, expanding access to communication infrastructure nationally should be a policy priority to ensure that health messages reach all populations. Our findings highlight the value of increased access to radio, television, and the internet, which serve as low-cost predisposing indicators for health information sharing across all settings in Nigeria. Policies should actively leverage these media to disseminate culturally appropriate maternal health messages, particularly targeting marginalised and underserved populations. Complementary approaches, including leveraging local community networks and engaging local champions, have proven effective in reaching marginalised populations [90,91]. For instance, lessons from the Abiye program (in Ondo state, Nigeria) demonstrate that community involvement and advocacy improved the uptake and sustainability of facility-based childbirth, helping to reduce rural–urban disparities [90,91]. Scaling such approaches nationally could strengthen health communication and reinforce behaviour change toward safer birth practices.

Third, addressing Andersen's environmental factors, particularly regional and rural–urban disparities, requires targeted structural action. In the North, interventions should combine infrastructure development, improved facility access, and enhanced security. In the South-South, policies should prioritise hard-to-reach areas such as riverine locations and address environmental degradation. Medium- to long-term priorities include investing in transportation networks, road infrastructure, and well-equipped healthcare centres across underserved regions. Short-term strategies, such as mobile health services and functional ambulances, can reach remote communities and enable timely transfers. Strengthening the health workforce and ensuring equitable distribution of skilled birth attendants remain essential. Yet improving access alone is insufficient; the quality of care, especially emergency obstetric services, must also improve [14,65]. The MSS offers a practical model, showing the effectiveness of deploying skilled personnel to underserved areas [15,19]. A revitalised MSS, integrated within UHC reforms, supported by sustainable financing, fair remuneration, and clear career pathways, would be feasible and equity-enhancing.

Fourth, targeted financial interventions are essential to address economic barriers that may contribute to home births in both rural and urban settings, including among middle-income households. Policies should expand and sustain maternal health subsidies to improve affordability and promote equitable access, for instance, through an expanded National Health Insurance Scheme (NHIS). Lessons from previous initiatives, such as the Abiye program and the CCT scheme [15–18,20,90,91], highlight the potential of demand-side financing to increase facility-based childbirth when adequately funded and monitored. Community-based financing models, including women's savings groups, micro-insurance, and voucher programs, can complement national strategies. Effective implementation requires strong governance, transparency, and integration within broader health financing reforms. The experience of the MSS, which suffered from limited collaboration and inconsistent funding, underscores the need for predictable financing and sustained political commitment to ensure program success [15,19].

Fifth, promoting maternal autonomy is vital, especially in rural areas. Engaging male partners and community networks in maternal health education can create supportive environments [85]. Programs that challenge restrictive gender norms and encourage shared decision-making are crucial for improving maternal health outcomes. Potential strategies could include participatory workshops, peer-support networks, and male engagement initiatives. These intervention efforts can help advance social justice by addressing power imbalances, enhancing maternal agency, and ensuring that marginalised women can make informed choices.

Finally, our findings call for regionally tailored, culturally sensitive strategies addressing Nigeria's distinct inequalities. In rural areas, policies should prioritise physical and informational access and women's empowerment [85]. Targeted investments in regions such as the North, where the Hausa-Fulani ethnic group is concentrated and home birth is high, are essential. In urban areas, focus should be on financial accessibility and culturally sensitive care, particularly for Muslim women who may face barriers related to privacy and gender preferences [86,87]. Faith-based approaches could help reduce home births among Muslims and other groups [88]. Across Nigeria, poor maternal outcomes are driven by systemic gaps undermining UHC. The Abiye Programme's success highlights that political will, sustainable financing, and community ownership are key [90,91]. Framing interventions through a social justice lens ensures prioritisation of the most marginalised: rural mothers with geographic or autonomy constraints, poor mothers, urban middle-wealth mothers, Muslim mothers in urban areas, younger urban mothers, and mothers with limited education, ANC, or health information access. Future research should evaluate the feasibility, scalability, and sustainability to ensure place-based innovations equitably reduce home births and maternal mortality.

## Strengths and limitations

This study demonstrates key strengths, including analysis at national, urban, and rural levels, which clarifies disparities in home birth practice in Nigeria. The findings offer evidence-based insights for policies to reduce home births in Nigeria and are generalizable to the larger population due to the nationally representative sample. However, there are limitations to consider. Firstly, data collection relied on mothers' recall ability; while it would be minimal given the substantial nature of the event, there appears to be a possibility of recall bias. Secondly, due to data limitations, the study did not include potential culture-related factors that could influence home births in Nigeria. However, we captured wide-ranging factors based on the NDHS dataset, providing comprehensive insights. The dataset used is over five years old and may not accurately reflect the current situation. Nonetheless, it remains the most recent edition and our findings serve as valuable evidence for further research on home births in Nigeria and other low- to middle-income countries. Finally, the data utilised were cross-sectional; hence, causality cannot be established, limiting the scope of our conclusions.

## Conclusion

Home birth remains widespread in Nigeria, particularly in rural areas and across the northern and South-South regions. Addressing this issue requires both universal and targeted strategies. Interventions should focus on common factors such as improving education, expanding access to antenatal care, increasing media and internet access, and alleviating financial barriers, while also tailoring solutions to the unique challenges in different contexts. In rural areas, interventions must prioritise improving physical access to healthcare (e.g., limiting distance barriers), enhancing women's autonomy for healthcare decision making, and investing in health infrastructure. In urban areas, our findings support a targeted focus on reducing economic vulnerability, including urban poverty and erosion of the middle class, prioritising youth-specific needs, and addressing religious or cultural influences. From an equity standpoint, responses must be locally grounded and ensure that all mothers, regardless of location, have access to safe, respectful, and high-quality maternal healthcare. Addressing home birth through these approaches is a critical step toward reducing Nigeria's persistently high maternal and neonatal mortality rates, underscoring the urgency of effective action.

# Acknowledgments

We are grateful to the DHS Program for providing access to the dataset used in this study. We also extend our sincere thanks to the mothers who participated, generously offering their time and valuable information. Additionally, we appreciate Oladimeji John Adewuyi for helping with preparing some of the Figures in this study.

# Author contributions

**Conceptualization:** Emmanuel O. Adewuyi, Mary I. Adewuyi.

**Data curation:** Emmanuel O. Adewuyi, Victory Olutuase.

**Formal analysis:** Emmanuel O. Adewuyi.

**Funding acquisition:** Emmanuel O. Adewuyi.

**Investigation:** Emmanuel O. Adewuyi, Asa Auta, Olumuyiwa Omonaiye, Mary I. Adewuyi, Kazeem Adefemi, Olumide A. Odeyemi, Yun Zhao, Gizachew A. Tessema, Gavin Pereira.

**Methodology:** Emmanuel O. Adewuyi, Asa Auta, Yun Zhao, Gizachew A. Tessema, Gavin Pereira.

**Project administration:** Emmanuel O. Adewuyi.

**Supervision:** Emmanuel O. Adewuyi.

**Validation:** Emmanuel O. Adewuyi, Asa Auta, Olumuyiwa Omonaiye, Victory Olutuase, Olumide A. Odeyemi, Yun Zhao, Gizachew A. Tessema, Gavin Pereira.

**Visualization:** Emmanuel O. Adewuyi.

**Writing – original draft:** Emmanuel O. Adewuyi, Olumuyiwa Omonaiye, Mary I. Adewuyi, Kazeem Adefemi, Gavin Pereira.

**Writing – review & editing:** Emmanuel O. Adewuyi, Asa Auta, Olumuyiwa Omonaiye, Mary I. Adewuyi, Victory Olutuase, Kazeem Adefemi, Olumide A. Odeyemi, Yun Zhao, Gizachew A Tessema, Gavin Pereira.

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
