## [Decision Letter · Decision Letter 0]

17 Sep 2025

Dear Dr. Adewuyi

Thank you for submitting your manuscript to PLOS ONE. After careful consideration, we feel that it has merit but does not fully meet PLOS ONE’s publication criteria as it currently stands. Therefore, we invite you to submit a revised version of the manuscript that addresses the points raised during the review process, by the Editor and the reviewers

We look forward to receiving your revised manuscript.

Kind regards,

Hannah Mafo Degge, Ph.D

Academic Editor

PLOS ONE

Journal Requirements:

“EOA was supported by the National Health and Medical Research Council (NHMRC; https://www.nhmrc.gov.au/) Investigator grants (GNT2025837). The funder has no role in the conduct of this study.”

In the figure caption of the copyrighted figure, please include the following text: “Reprinted from [ref] under a CC BY license, with permission from [name of publisher], original copyright [original copyright year].

Additional Editor Comments :

Introduction

This section could benefit with some critical appraisal

e.g Line 86-87; you mentioned the MSS and conditional cash transfers; however MSS was not sustained and conditional cash transfer was not widespread; provide a critical analysis of this and its probable contribution to differential outcomes across states/regions

line 97- home birth in Nigeria and LMICs; home birth in Nigeria is currently mostly TBA provided or unsupervised ‘SBA’, this is the core of your paper, hence will be useful to note this, and explore the whys of the contributions of these two ‘providers’ to poor maternal outcomes, including briefly touching why women will opt for a home birth based on past literature ( be careful when doing this in not giving too much away) as this is the focus of your research

line 161 –it will be useful to provide specific maternal health initiative that align with global health rather than just saying ‘some initiatives…

Discussion

You mentioned using the theoretical frameworks Andersen’s Behavioural Model of Healthcare- this has not been integrated or applied explicitly in your discussion; will be beneficial to do this and not just limit to using it in identifying your explanatory variable. Consider how you can apply this to strengthen your discussion

line 375-376- decline mentioned is from which data, kindly indicate what, and/or is it the previous DHS and why might this decrease have happened?

Lines 380-382; similar to line 97-99 your introduction- possible repetition?

Line 384-385, explore further why urban residence alone does not guarantee for that region

Line 389- what constrained options? This is too vague, kindly provide details/examples

Line 396- you wrote “This premise also supports the urgent need for a social justice approach—one that goes beyond individual-level interventions to confront systemic barriers.”

Elaborate on what you mean by social justice approach in the context of your study

Lines 436-439; you wrote “birth among women in the middle wealth category in urban settings only, possibly reflect the reality of urban poverty, where families live in informal settlements or slums, underserved by health infrastructure, or the erosion of the middle class in urban Nigeria”

This is too long, kindly split and rephrase to improve clarity

Lines 464-467- some critical analysis required in your suggestion on digital inclusion and integration of internet access into maternal health; will this not widen the existing health inequality, considering digital inclusion means access to smart phones- in the current climate of lack of basic infrastructure including electricity, poor internet connectivity, high cost of internet services; is this feasible across all states and all regions?

Kindly note it is not just about suggesting policy that promote digital inclusion, what about the enabling factors to allow for the implementation of the policy, provide a crtiical discourse on this point

Line 475-477 you wrote “The markedly higher odds of home birth in the rural North-West and South-South may also exemplify how rurality and regional deprivation intersect to perpetuate maternal health disparities”.

Kindly expand on what you mean – noting the keen differences in North and South of Nigeria

Policy and programmatic implications

The way you have listed them all out in this section, your implications all appear like the typical suggestions. Going forward in rewriting this section; a critical consideration of feasibility will be beneficial in presenting this. Also, it is good for you to focus on innovative means (cost effective, community driven initiatives) that could be employed:

Consider the MSS that was started and abandoned due to number of reasons, including lack of collaboration from all partners. How can MSS be revamped and resuscitated? How do we address the user fee issue that limits access? What innovative community driven financial scheme can be used by women to address the use of user fee?

In considering the systemic/structural issues, what can you learn from the Abiye programme in Ondo state, why did it succeed and why has it gone down? What take home point can you use from this in addressing the systemic/structural issues

Still in line with above, controlling all other factors (like culture, ethnicity, or religion), the bottom line to the poor maternal outcome is the absence of good maternal health care system(systemic/strucutral factors) that meets the requirement of UHC (hence the need to look at the Abiye as mentioned above) what lessons can we learn and apply

Finally, since you used the social justice lens, kindly organise this section along that line so it does not appear like a ‘textbook prescription’,

Minor corrections

Proofread and review for typo errors. E.g check your acknowledgement line 562, what is figs?

Kindly refrain from the use of dash (-) in academic writing it sets an informal tone, and it is associated with AI generated work

Reviewers' comments:

Reviewer's Responses to Questions

**Comments to the Author**

1. Is the manuscript technically sound, and do the data support the conclusions?

Reviewer #1: Yes

Reviewer #2: Yes

Reviewer #3: Yes

Reviewer #4: Yes

2. Has the statistical analysis been performed appropriately and rigorously?

Reviewer #1: Yes

Reviewer #2: Yes

Reviewer #3: Yes

Reviewer #4: Yes

3. Have the authors made all data underlying the findings in their manuscript fully available?

Reviewer #1: Yes

Reviewer #2: Yes

Reviewer #3: Yes

Reviewer #4: No

4. Is the manuscript presented in an intelligible fashion and written in standard English?

Reviewer #1: Yes

Reviewer #2: Yes

Reviewer #3: Yes

Reviewer #4: Yes

Reviewer #1: 1. Since you are using the Andersen model, kindly provide a brief description of the healthcare facility distribution in Nigeria (e.g., availability across states).

2. Line 185: Ethical approval was granted by the Nigerian National Health Research Ethics Committee, could you provide the ethical approval reference number from the Nigerian National Health Research Ethics Committee for this study?

3. Table 1, the wording used for the enabling factors included was quite informal:

"Distance to health facility (big problem/not a big problem)"

"Permission to visit health facility (big problem/not a big problem)"

"Getting money for health services (big problem/not a big problem)".

Suggested revisions:

Difficulty due to distance (Yes/No)

Decision-making autonomy (Independent/Requires permission)

Affordability issue (Yes/No)

4. Table 2: Would you consider rearranging the order of the variables based on domain in Andersen model (to be similar to the order in Table 1)?

5. Table 2: Please consider adding a p-value column for clarity.

6. Line 454: Higher-parity women have higher odds of home birth; Line 502, Younger women have higher odds of home birth. However, younger women usually have lower parity, while higher-parity mothers are older. Could you reconcile or further explain these findings?

7. Line 500: You noted higher odds of home births among Muslim mothers. Could you provide supporting evidence on Muslim perspectives toward maternal health services in Nigeria, since you suggest culturally sensitive services?

Reviewer #2: as far as i searched the study is original research and not published else where.

starting from the research topic it is impactful and pertinent issue but I have doubt on its novelty. since it was done before based on 2013 Nigeria demographic and health survey , it needs pertinent justification.

even if conclusion lacks some reference data but recommendations were stated based on the finding data

better to select pertinent keywords in the abstract part

introduction part is well written but significant impact of home birth not mentioned briefly

add reference at the end of 1st paragraph of introduction part line number 87-89

in Methods part how informed consent was applicable from secondary data? how do you get the participant? line number 185-187)

operational definition should included in the method part to clarify some variables like rich house hold and poor house hold

discussion part is well written and appreciated

Reviewer #3: General remarks about the manuscript

This manuscript tackles a crucial public health issue—the ongoing challenge of home births in Nigeria and the disparities that exist between rural and urban regions. Utilizing a substantial, national dataset to examine these variations is a notable strength of the study. The emphasis on identifying related factors is particularly pertinent for shaping public health policies and interventions. The manuscript is well-articulated, and the research question is clearly defined. My following observations are aimed at enhancing the paper and preparing it for publication.

Specific comments

1. In result part of your abstract add confidence interval and p- value for all prevalence value or remove all, it is better to be similar. And also better to add p- Value and CI for the AOR value of significant variables and home birth.

2. Maternal morbidity and mortality, which are the results of home delivery, are discussed in the first paragraph of your introduction. But your first paragraph of introduction part is better to deal about home birth and the consequence of it in globally. Therefore, it is preferable to eliminate the first paragraph or reorganize the introduction's sequence.

3. In your methodology part, the total 41,821 mothers were surveyed, but you take only 21,512 mothers why? Or how can you select those sampled mothers?

4. In your analysis section, you assess the multicollinearity assumption, which is good, but what about other other logistic regression assumptions like outlier detection, the Hosmer and Lemeshow test, and the independence of observations?

5. In you discussion part, the manuscript title highlights a comparative analysis between rural and urban settings. The discussion should not only present these differences but also delve into the potential underlying reasons for them. For instance, are the rural-urban disparities due to differences in healthcare infrastructure, socioeconomic status, cultural beliefs, or a combination of these factors? A deeper interpretation would significantly strengthen the findings.

6. Ensure consistent use of terminology throughout the manuscript. For example, "home birth" and "childbirth outside health facilities" appear to be used interchangeably. A single, clear definition at the beginning of the manuscript would be beneficial.

Reviewer #4: Thank you for the opportunity to review this excellent and important paper. Overall, I think it is well done. I do have some concerns that this paper was published with data from Nigeria without any authors who have an affiliation in Nigeria. I understand that maybe there are connections to Nigeria that I am not aware of but would like it to be addressed and explicitly stated if there is. If not, I would like to see this addressed in the limitations or potentially adding a reflexivity statement https://gh.bmj.com/content/9/1/e014743.

Line 81-specify "maternal and neonatal" mortality rates

Line 85-it is not just extremely high but the highest of any country in the world, I think this should be added.

Line 88-Nonetheless, maternal and neonatal mortality remain unacceptably high in Nigeria, with the continued prevalence of home births emerging as a likely key contributing factor. -this should be referenced.

line 99-lack of skilled birth attendants is also a significant factor here based on data and should be specifically called out.

line 101- Hence, ensuring facility-based births with access to emergency care remains essential to reducing

these deaths. you should add and access to skilled birth attendants here

I really love this dual approach -This dual lens shifts the focus beyond individual behaviours to systemic

137 constraints in maternal healthcare access.

the methods are sound and well done

Line 391-also women not be empowered also likely plays a role here and should be called out here as well as later when you discuss it more

I would also add a line in the conclusions tying back in the maternal and newborn mortality to remind the readers why it is important to address home births.

Very nice job with this paper

**Do you want your identity to be public for this peer review?** For information about this choice, including consent withdrawal, please see our Privacy Policy

Reviewer #1: No

Reviewer #2: **Yes: ** Lubaba Ahmed Mohammed

Reviewer #3: No

Reviewer #4: No

---

## [Author Response · Author response to Decision Letter 1]

5 Oct 2025

Dear Editor,

We appreciate the meticulous review of our manuscript and the insightful comments and suggestions of the editor and reviewers. We have thoroughly considered these comments and revised our manuscript accordingly. Below, we present our point-by-point responses to the comments and suggestions. Line references refer to those in the tracked revised manuscript, except where otherwise noted.

In addition to the comments and recommendations of the reviewers:

1. We made a minor correction to the title, which avoids repetition of ‘based’.

New title:

Home birth and associated factors in Nigeria: a comparative study of rural and urban settings—analysis of national population-based data

2. We made minor corrections throughout to improve readability.

3. We proofread and spell-checked our manuscript.

Editor Comments

Journal Requirements:

Comment 1. Please ensure that your manuscript meets PLOS ONE's style requirements, including those for file naming. The PLOS ONE style templates can be found at

Response: We thank the editor for the comment and guidance. We confirm that we have reviewed the links provided and have followed the templates from the outset (initial submission) of our manuscript. Additionally, we ensured that the naming of files complies with the requirements.

Comment 2. Thank you for stating in your Funding Statement:

“EOA was supported by the National Health and Medical Research Council (NHMRC; https://www.nhmrc.gov.au/) Investigator grants (GNT2025837). The funder has no role in the conduct of this study.” Please provide an amended statement that declares *all* the funding or sources of support (whether external or internal to your organization) received during this study, as detailed online in our guide for authors at http://journals.plos.org/plosone/s/submit-now. Please also include the statement “There was no additional external funding received for this study.” in your updated Funding Statement. Please include your amended Funding Statement within your cover letter. We will change the online submission form on your behalf.

Response: We thank the editor for guidance regarding the Funding Statement, and have made the requested correction.

Amended Funding Statement: EOA was supported by the National Health and Medical Research Council (NHMRC; https://www.nhmrc.gov.au/) Investigator Grant (GNT2025837). The funder had no role in the conduct of this study. There was no additional external funding received for this study.

Comment 3. We note that you have indicated that there are restrictions to data sharing for this study. For studies involving human research participant data or other sensitive data, we encourage authors to share de-identified or anonymized data. However, when data cannot be publicly shared for ethical reasons, we allow authors to make their data sets available upon request. For information on unacceptable data access restrictions, please see http://journals.plos.org/plosone/s/data-availability#loc-unacceptable-data-access-restrictions.

Response: Thank you for your guidance regarding the data availability statement. The data used in this study are from the Demographic and Health Surveys (DHS) Program. These data are owned and managed by the DHS Program, which imposes restrictions on their public sharing. As such, we are not permitted to upload or distribute the datasets directly. Access is granted by the DHS Program after registration and approval of a research request through their website (https://dhsprogram.com). Approval is determined by the DHS Program, which serves as the data custodian, to ensure that use complies with its ethical and legal requirements. Researchers wishing to access the data may submit a request to the DHS Program, which reviews applications and provides de-identified datasets under strict terms of use.

Amended Data Availability Statement: The data that support the findings of this study are available from the Demographic and Health Surveys (DHS) Program, but restrictions apply to their availability. These data were used under license for the current study and are not publicly shareable. Data are, however, freely available upon reasonable request from the DHS Program (https://dhsprogram.com/data/Access-Instructions.cfm) following their approval process. The authors did not have any special access privileges, and others may access the data in the same manner.

Comment 4. We note that Figure 1 in your submission contain [map/satellite] images which may be copyrighted. All PLOS content is published under the Creative Commons Attribution License (CC BY 4.0), which means that the manuscript, images, and Supporting Information files will be freely available online, and any third party is permitted to access, download, copy, distribute, and use these materials in any way, even commercially, with proper attribution. For these reasons, we cannot publish previously copyrighted maps or satellite images created using proprietary data, such as Google software (Google Maps, Street View, and Earth). For more information, see our copyright guidelines: http://journals.plos.org/plosone/s/licenses-and-copyright.

Response: We confirm that Figure 1 was entirely generated by the authors using R and open-source data. Specifically, the figure was created using the rnaturalearth and rnaturalearthdata packages to obtain publicly available shapefiles for Nigeria, along with ggplot2, sf, dplyr, and ggrepel for visualisation and labelling. No copyrighted maps, satellite imagery, or proprietary software (e.g., Google Maps, Google Earth) were used in generating this figure. As all components of the figure were created from public-domain data and original code, Figure 1 can be published freely without restriction.

To ensure transparency and reproducibility, we have made the R code used to generate this figure publicly available online at a GitHub page: ‘https://github.com/actright1/Nigeria-Geopolitical-Map’, which can be freely used by others.

Also, we have included a statement in-text: ‘Note: Map of Nigeria showing states and geopolitical zones. The map was generated by the authors using R with data from the rnaturalearth and rnaturalearthdata packages (public domain). A freely accessible script for generating the map is available at https://github.com/actright1/Nigeria-Geopolitical-Map (lines 233 - 236).

Comment 5. Please review your reference list to ensure that it is complete and correct. If you have cited papers that have been retracted, please include the rationale for doing so in the manuscript text, or remove these references and replace them with relevant current references. Any changes to the reference list should be mentioned in the rebuttal letter that accompanies your revised manuscript. If you need to cite a retracted article, indicate the article’s retracted status in the References list and also include a citation and full reference for the retraction notice.

Response: We have checked to ensure all references are complete and correct. To the best of our knowledge, there are no retracted studies cited in our work.

Additional Editor Comments :

Comment: This section could benefit with some critical appraisal

e.g Line 86-87; you mentioned the MSS and conditional cash transfers; however MSS was not sustained and conditional cash transfer was not widespread; provide a critical analysis of this and its probable contribution to differential outcomes across states/regions.

Response: We thank the editor for this suggestion. In the revised manuscript, we included an appraisal of both the MSS and CCT programmes. We also included the Basic Health Care Provision Fund (BHCPF).

‘…While these programmes yielded some benefits, their overall implementation was limited. The MSS faced challenges with midwife retention, lack of ongoing training, inconsistent funding and lack of commitment from subnational governments [1, 2]. Similarly, CCT schemes were small-scale, intermittent, and in some cases withdrawn or delayed, undermining trust and continuity of care [3, 4]. The BHCPF, though promising in design [5-7], has struggled with underfunding, delays in fund disbursement, capacity gaps, weaknesses in supervision and coordination, and uneven rollout. These implementation challenges may partly explain the persistence of home births, a potential contributor to Nigeria’s high maternal and neonatal mortality.’ (101 – 108).

Comment: line 97- home birth in Nigeria and LMICs; home birth in Nigeria is currently mostly TBA provided or unsupervised ‘SBA’, this is the core of your paper, hence will be useful to note this, and explore the whys of the contributions of these two ‘providers’ to poor maternal outcomes, including briefly touching why women will opt for a home birth based on past literature ( be careful when doing this in not giving too much away) as this is the focus of your research

Response: We clarified in the Introduction that most home births in Nigeria are attended by unskilled traditional birth attendants, while some occur without any attendant. We note that the focus of the present study is on estimating the prevalence of home births and examining associated factors across overall, rural and urban residences. Analysis of provider contributions to maternal outcomes is not the main focus. Brief contextual information (to focus on our topic and avoid making the writing too dense) from prior literature has been included to situate the study, without pre-empting the study findings, as provided in the paragraph below:

‘…Yet in Nigeria, home births remain common [8-14], often attended by unskilled traditional birth attendants or, in some cases, without any attendant [15-17]. This premise may reflect both structural inequities and sociocultural practices, highlighting the need for a deeper understanding of the possible drivers of home birth in the country.’ (lines 119 – 122).

Comment: line 161 –it will be useful to provide specific maternal health initiative that align with global health rather than just saying ‘some initiatives…

Response: We have revised the text to specify the maternal health initiatives implemented by the Nigerian government that align with global health priorities. We now mention the MSS and CCTs, which were designed to improve access to maternal and neonatal healthcare and service utilisation [1, 3, 4, 18].

‘To improve maternal and neonatal health and align with global health goals, the Nigerian government has implemented initiatives such as the MSS, CCTs and BHCPF [1, 3-7, 18]. Despite these efforts, the country continues to experience poor maternal and neonatal health outcomes. Home birth is a potential contributor, and national data may obscure within-population disparities [19-21]; hence the approach in this study.’ (lines 266 – 271).

Discussion

Comment: You mentioned using the theoretical frameworks Andersen’s Behavioural Model of Healthcare- this has not been integrated or applied explicitly in your discussion; will be beneficial to do this and not just limit to using it in identifying your explanatory variable. Consider how you can apply this to strengthen your discussion.

Response: We thank the editor for this valuable suggestion. We have now reflected Andersen’s model within the discussion. Beyond informing explanatory variables, we used the model’s domains: predisposing, enabling, and environmental factors, to briefly contextualise our findings. For instance, the region of residence, an environmental factor, was highlighted as potentially shaping the availability, accessibility, and utilisation of maternal health services, which underpins the observed regional and rural–urban disparities. Similarly, maternal and partner’s education, household wealth, ethnicity, parity, and media exposure were discussed as predisposing factors influencing mothers’ odds of home birth. Financial barriers, distance to facilities, and ANC utilisation were highlighted as enabling factors.

Together, these domains allowed us to interpret the observed patterns within a coherent theoretical framework. Importantly, we examined them not only through Andersen’s model but also through the broader lens of social determinants of health and social justice. This combined framing underscored how structural and systemic inequities may contribute to driving home birth in Nigeria, and highlighted the urgency of prioritising the most marginalised mothers in policy and practice. The approach was also reflected in our policy recommendations.

Comment: line 375-376- decline mentioned is from which data, kindly indicate what, and/or is it the previous DHS and why might this decrease have happened?

Response: We have clarified that the comparison figures are from the 2013 NDHS (with appropriate citation) and added a note that the modest decline may be due to the impact of interventions introduced in the past decades, while recognising that disparities persist and home birth remains prevalent. We prioritised being concise, so it does not become too dense, as we already have a lengthy discussion section.

‘Although the figures represent a modest decline from the 2013 estimates (63% nationally, 78% rural, and 38% urban) [10, 14], the burden remains among the highest globally, with the national figure exceeding estimates in many African countries and the global average of 28% [22-25]. The modest decline observed in our study may reflect the impact of interventions implemented over the past decades in Nigeria [1, 3-7, 18]. (lines 547 – 552)

Despite these marginal improvements, the high burden of home births in Nigeria is concerning. The observed reductions fall short of global targets, measured by the proportion of childbirth attended by skilled personnel, a key indicator for achieving SDG 3.1 [26, 27]. The continued high prevalence is notable, as home births typically occur in Nigeria without access to emergency obstetric care or, in some cases, without anyone in attendance [15, 16].‘ (lines 556 – 562)

Comment: Lines 380-382; similar to line 97-99 your introduction- possible repetition?

Response: We note that the introduction and discussion serve distinct purposes. In the introduction (lines 97–102 referenced from the previous submission), we present the general public health challenge of home births in LMICs and the associated risks. In the discussion (lines 380–382 of the previous submission), we applied the idea to our study findings by highlighting how the factors are associated in the Nigerian context (based on our findings). In the revised manuscript, we have double-checked to ensure everything harmonise well.

Comment: Line 384-385, explore further why urban residence alone does not guarantee for that region

Response: We have revised the discussion to concisely clarify why urban residence in the North

---

## [Editor Report · Decision Letter 1]

19 Oct 2025

Dear Dr. Adewuyi,

Thank you for revising and resubmitting your manuscript to PLOS ONE. After careful consideration, we feel that it has merit but does not fully meet PLOS ONE’s publication criteria as it currently stands. And there are few minor revisions required to achieve this standard. Therefore, we invite you to submit a revised version of the manuscript that addresses the points raised during the review process.

**ACADEMIC EDITOR: **

Address the following

1. Repetition observed in lines 91-94 and 193-194

"91 In response to this ongoing challenge, the Nigerian government has introduced several initiatives

92 aimed at increasing access to skilled care and improving maternal and newborn outcomes. These

93 initiatives include the Midwives Service Scheme (MSS), conditional cash transfers (CCTs), and the Basic

94 Health Care Provision Fund (BHCPF) [15-22]".

"193 To improve maternal and neonatal health and align with global health goals, the Nigerian government

194 has implemented initiatives such as the MSS, CCTs and BHCPF [15-18, 20-22]."

Consider  merging this last paragraph under Study Setting with the paragraph above it and reframing lines 193-194 to read something like this:

*"Despite the Nigerian government’s efforts to improve maternal and neonatal health in line with global health goals, through initiatives such as the MSS, CCTs, and BHCPF mentioned earlier; poor maternal and neonatal health outcomes continue to persist"*

2. Sentence in line 235-236 should be in past tense, amend accordingly

235 We adopt Andersen’s Behavioural Model as the conceptual framework and utilise the model in

236 selecting explanatory variables

We look forward to receiving your revised manuscript.

Kind regards,

Hannah Mafo Degge, Ph.D

Academic Editor

PLOS ONE

Journal Requirements:

Editor's comments

SEE ABOVE

---

## [Author Response · Author response to Decision Letter 2]

6 Nov 2025

Response to Editor and Comments

We appreciate the editor’s constructive comments and review of our manuscript. We have carefully considered each point and revised the manuscript accordingly. A detailed, point-by-point response is provided below. We have also taken this opportunity to conduct a thorough proofreading of the entire manuscript to enhance clarity and consistency.

Comment 1:

Repetition observed in lines 91–94 and 193–194. Consider merging the last paragraph under Study Setting with the paragraph above it and reframing lines 193–194 to read:

“Despite the Nigerian government’s efforts to improve maternal and neonatal health in line with global health goals, through initiatives such as the MSS, CCTs, and BHCPF mentioned earlier; poor maternal and neonatal health outcomes continue to persist.”

Response:

We appreciate this observation. The repetition has been addressed. The paragraph under Study Setting has been merged with the preceding one, and lines 193–194 have been revised following the suggested phrasing to improve coherence and eliminate redundancy.

‘Despite the Nigerian government’s efforts to improve maternal and neonatal health in line with global health goals, through initiatives such as the MSS, CCTs, and BHCPF [1-7], mentioned earlier, poor maternal and neonatal health outcomes continue to persist.’ (lines 201 – 203)

Comment 2:

Sentence in lines 235–236 should be in past tense.

“We adopt Andersen’s Behavioural Model as the conceptual framework and utilise the model in selecting explanatory variables.”

Response:

Thank you for this useful suggestion. The sentence has been revised to the past tense for consistency with the rest of the manuscript:

‘We adopted Andersen’s Behavioural Model as the conceptual framework and utilised the model in selecting explanatory variables.’ (lines 244 – 245)

Lastly, we have carried out a careful proofreading of the whole manuscript to enhance clarity and consistency.

References

1. Chukwuma, J.N., Implementing Health Policy in Nigeria: The Basic Health Care Provision Fund as a Catalyst for Achieving Universal Health Coverage? Development and Change, 2023. 54(6): p. 1480-1503.

2. Igbokwe, U., et al., Evaluating the implementation of the National Primary Health Care Development Agency (NPHCDA) gateway for the Basic Healthcare Provision Fund (BHCPF) across six Northern states in Nigeria. BMC Health Services Research, 2024. 24(1): p. 1404.

3. Ibrahim, Z.A., et al., Influence of Basic Health Care Provision Fund in improving primary Health Care in Kano state, a descriptive cross-sectional study. BMC Health Services Research, 2023. 23(1): p. 885.

4. Abimbola, S., et al., The Midwives Service Scheme in Nigeria. PLOS Medicine, 2012. 9(5): p. e1001211.

5. Federal Ministry of Health. Saving newborn lives in Nigeria: Newborn health in the context of the Integrated Maternal, Newborn and Child Health Strategy. 2011; 2nd Edition:[Available from: https://dc.sourceafrica.net/documents/120214-NEWBORN-HEALTH-in-the-Context-of-the-Integrated.html.

6. Okoli, U., et al., Conditional cash transfer schemes in Nigeria: potential gains for maternal and child health service uptake in a national pilot programme. BMC pregnancy and childbirth, 2014. 14: p. 1-13.

7. Ezenwaka, U., et al., Influence of Conditional Cash Transfers on the Uptake of Maternal and Child Health Services in Nigeria: Insights From a Mixed-Methods Study. Frontiers in Public Health, 2021. 9.

---

## [Editor Report · Decision Letter 2]

9 Nov 2025

Home birth and associated factors in Nigeria: a comparative study of rural and urban settings—analysis of national population-based data

PONE-D-25-25157R2

Dear Dr. Adewuyi,

We’re pleased to inform you that your manuscript has been judged scientifically suitable for publication and will be formally accepted for publication once it meets all outstanding technical requirements.

Kind regards,

Hannah Mafo Degge, Ph.D

Academic Editor

PLOS ONE

---

## [Editor Report · Acceptance letter]

PONE-D-25-25157R2

PLOS ONE

Dear Dr. Adewuyi,

I'm pleased to inform you that your manuscript has been deemed suitable for publication in PLOS ONE. Congratulations! Your manuscript is now being handed over to our production team.

Kind regards,

on behalf of

Dr. Hannah Mafo Degge

Academic Editor

PLOS ONE